# Battery electric vehicles show the lowest carbon footprints among passenger cars across 1.5–3.0 °C energy decarbonisation pathways

Joris Šimaitis[1] ✉, Rick Lupton [1], Christopher Vagg [2], Isabela Butnar [3], Romain Sacchi [4] & Stephen Allen [1]

Passenger car carbon footprints are highly sensitive to future energy systems, a factor often overlooked in life cycle assessment. We use a time-dependent prospective life cycle assessment to enhance carbon footprints under four 1.5–3.0 °C decarbonisation pathways for electricity, fuel, and hydrogen from an energy-based integrated assessment model. Across 5000 comparative cases, battery electric vehicles consistently have the lowest carbon footprints compared to hybrid, plug-in hybrid, and fuel-cell vehicles. For example, battery electric vehicles show an average 32 to 47% lower footprint than hybrid combustion in 3.0 °C and 1.5 °C climate-compatible futures, respectively. This is driven by greater projected decarbonisation of electricity compared to fossil-dominated fuels and hydrogen. Battery electric vehicles meaningfully retain their advantage for mileages over 100,000 km, even in regions with carbon-intensive electricity since these are anticipated to decarbonise the most. Although our study supports battery electric vehicles as the most reliable climate-mitigation option for passenger cars, reducing their high manufacturing footprint remains important.

Road transport energy use contributes 12% to global greenhouse gas (GHG) emissions, mainly due to fossil-fuel use in internal combustion engine vehicles (ICEVs)[1]. To reduce emissions, the adoption of battery electric vehicles (BEV) has risen, with BEVs making up nearly 10% of global vehicle sales in 2021—four times the 2019 market share[2].

Many life cycle assessment (LCA) studies indicate that BEVs generally have a lower life-cycle global warming potential (GWP) (GWP, hereafter referred to as "carbon or carbon footprint") than ICEVs across most regions[3–5]. While BEVs have a higher embodied carbon due to energy-intensive battery production, this is outweighed by their efficient powertrain which utilises electricity during operation that is less carbon-intensive compared to fossil-fuel supply and combustion in ICEVs[6,7]. However, in regions with high-carbon electricity, such as India or China (CHI), BEVs have been shown to have higher carbon footprints than ICEVs[8,9]. Although it is argued BEV use phase impacts will continue to decrease over time as electricity grids continue to decarbonise[10]. However, counterpoints suggest that considering low-carbon fuels adoption could make combustion and hybrid powertrains competitive or even preferable[11–13]. In any case, the future energy mix, along with the magnitude and rate of decarbonisation, is uncertain but vital in determining passenger car carbon footprints.

However, most passenger car LCA studies do not capture how future energy evolves over a vehicle lifetime, modelling the life cycle inventory using historical data, assuming manufacturing, use, and end-of-life occur simultaneously (e.g., year 0)[14]. In reality, manufacturing happens in year 0, the use phase may span years 1 to 15, and end-of-life may occur in year 16. Using year 0 electricity and fuel data across all life-cycle stages negates future changes in energy systems over the next 16 years. Consequently, life-cycle stages that extend into the future are not accurately depicted, such as vehicle use, a dominant carbon hotspot. Therefore, life-cycle stages must incorporate and evaluate future energy system transitions and adjust impacts based on when they are expected to occur.

In recent years, prospective LCA (pLCA) using future scenarios from integrated assessment models (IAMs) has emerged that can help address these challenges[15,16]. IAMs are widely recognised climate mitigation tools for exploring future energy transitions through socioeconomic and cost-optimisation modelling[17], and are closely related and complimentary to

[1]Institute of Sustainability and Climate Change, University of Bath, Bath, UK. [2]Institute for Advanced Automotive Propulsions Systems, University of Bath, Bath, UK. [3]Institute for Sustainable Resources, University College London, London, UK. [4]Laboratory for Energy Systems Analysis, Centers for Energy and Environmental Sciences and Nuclear Engineering and Sciences, Paul Scherrer Institute, Villigen, Switzerland. ✉e-mail: js3700@bath.ac.uk

energy system models[18]. In particular, the introduction of "Premise"[19] has allowed IAM future scenarios to be integrated into the ecoinvent[20] LCA database. Premise maps IAM variables to LCA activities and generates prospective versions of the LCA database by adjusting technologies' penetration share, efficiency and emission factors for a specific scenario and year, considering regionalised sectoral changes related to electricity, fuels, cement, steel and more. This has enabled pLCA studies to explore the future environmental impacts of technologies such as batteries[21], hydrogen[22], and cement[23]. However, there have been few pLCA studies that utilise the IAM approach for passenger vehicle comparisons[6,24,25].

Mendoza Beltran et al.[24] were the first to apply this approach using future scenarios from the IMAGE IAM to compare the carbon footprints of BEVs and ICEVs. They showed that accounting for the future electricity mix found considerably favourable outcomes for BEVs. However, the study was primarily a proof of concept and had a simplistic powertrain analysis relying on default, outdated ecoinvent 3.3 vehicle inventories. Cox et al.[25] expanded on this by developing comprehensive vehicle inventories that compared not only BEVs and ICEVs but also mild hybrid combustion (HEVs), plug-in hybrid (PHEVs), and hydrogen fuel cell vehicles (FCEVs) across various configurations. Their pLCA findings reinforced previous conclusions that BEVs, in addition to FCEVs, offer the lowest present and future carbon footprints. Sacchi et al.[6] then leveraged these inventories for pLCA of passenger cars using the REMIND IAM in a multi-regional scenario assessment, confirming that BEVs currently outperform other powertrains in most regions, with their advantage growing as future electricity grids decarbonise.

However, these studies have key limitations. First, beyond a limited regional analysis, Mendoza Beltran et al.[24] and Cox et al.[25] focused solely on future electricity transformations from IMAGE, disproportionately favouring BEVs while neglecting broader energy system transitions in liquid fuels and hydrogen, key factors for non-electric powertrains. Moreover, they assessed vehicle impacts as static snapshots, assuming all life cycle stages occur simultaneously within either present or future systems, without precisely accounting for their dynamic, time-adjusted distribution over a vehicle's lifespan. Sacchi et al.[6] further tackled this by integrating REMIND's broader energy system changes, applying time-adjusted modelling, and providing multi-regional insights. But a critical pitfall was the inconsistency in the future scenario application. REMIND scenarios were used to transform the ecoinvent 3.8 background system (e.g., upstream and downstream supply chains). However, the foreground system, such as the electricity mix supply, was derived using only one future scenario per region, independent from the IAM (e.g., ENTSO-E) which also only considered substantial electricity decarbonisation, inherently favouring BEVs. Therefore, limited future scenarios were considered in addition to the background and foreground systems representing inconsistent futures. This compromised the methodological robustness and conclusions for passenger car carbon footprints. Finally, expanding pLCA studies beyond IMAGE and REMIND is essential, as different IAMs present varied technology pathways for decarbonisation and show differences in both the rates and magnitudes of decarbonisation for equivalent scenarios[26]. Therefore, it is crucial to understand pLCA results from a variety of IAMs as adopted by the IPCC[27] in evaluating climate mitigation pathways.

This work builds on previous studies to improve the methodological consistency of applying multiple IAM future scenarios and enhance the robustness of passenger car carbon footprints while holistically capturing broader energy system transitions. First, we introduce four new scenarios from TIAM-UCL into Premise to expand beyond the results available from IMAGE and REMIND, systematically covering 1.5–3.0 °C climate mitigation pathways for electricity, diesel, hydrogen, steel and more[26]. TIAM-UCL is particularly advantageous as it has an extensive focus on energy system technologies and an enhanced representation of fossil systems for enhanced depiction of energy transitions[28–30]. Next, we compile various vehicle configurations into a pLCA model and use a Monte Carlo (MC) approach to sample 5000 combinations for BEVs, FCEVs, HEVs, and PHEVs across seven sizes (e.g., from small to large SUVs), 16 regions (e.g., China, Western

Europe (WEU), and the United Kingdom), 100,000–300,000 km driven distances, 10–20 years lifetimes, and the four decarbonisation pathways. The pLCA then consistently time-adjusts production, use, and end-of-life stages to occur in the appropriate present and future periods based on the IAM future scenario. Finally, we comparatively evaluate various passenger car carbon footprints of current and future cases and utilise a Global Sensitivity Analysis (GSA) to quantify the influence of future scenarios and vehicle configurations.

## Results

### Future energy supply
Figure 1 presents the four global energy scenarios from TIAM-UCL for electricity, diesel, and hydrogen supply for passenger vehicles. The scenarios include "No climate action", which is aligned with limiting global warming to 3 °C and slow decarbonisation, and "Baseline (National Determined Contributions (NDCs))", which is the current pathway for meeting 2.0–2.5 °C with increased efforts for decarbonisation.

Paris Agreement-aligned pathways that show the most accelerated decarbonisation efforts are "Ambitious (2.0 °C)" and "Very ambitious (1.5 °C)". Figure 1a highlights the current dominance of coal and natural gas in the electricity mix. Future decarbonisation scenarios show their phase-out, and uptake of renewable energy, particularly solar and wind. Figure 1b shows that diesel production remains largely fossil-based but is expected to be phased out with future decarbonisation efforts, replaced with some synthetic and biomass-derived alternatives. Figure 1c illustrates that hydrogen production is also currently fossil-based, but is projected to shift towards cleaner sources, such as electrolysis and bioenergy with carbon capture and storage (BECCS), if decarbonisation efforts accelerate. More broadly, the Paris Agreement-compatible scenarios expect substantial power growth for electricity and hydrogen to decarbonise a wide range of sectors. The large differences between scenarios, which develop well within the lifetime of today's vehicles, demonstrate the importance of taking a prospective life-cycle approach. It is important to note that while absolute production values provide valuable context for the energy transition, LCA results are determined by the annual percentage mix of energy types within production volumes.

### Comparative carbon footprints
Figure 2 presents the pLCA of comparative life-cycle carbon footprints of passenger vehicles produced in 2025, showing pairwise comparisons across various vehicles and sampled input combinations, together with the mean result for each future scenario derived. Powertrains are compared for each sample for a consistent set of conditions (e.g., each point in Fig. 2a compares the carbon footprint of a BEV vs. HEV, for the same mileage, region, etc).

Passenger vehicle carbon footprints calculated using the time-adjusted pLCA, which accounts for future decarbonisation scenarios, differ considerably from LCA results using no future scenario. For instance, Fig. 2a shows that BEVs have a lower carbon footprint than HEVs in 99% of cases (data points are concentrated in the HEV area of the chart), with a mean difference ranging from 32% in the "No climate action" scenario (206 vs. 302 g $CO_2$e per km) to 47% in the "Very ambitious (1.5 °C)" scenario (142 vs. 270 g $CO_2$e per km). This contrasts with the "No scenario" (typical LCA) results, which find that BEVs have a smaller mean difference by 24% (219 vs. 289 g $CO_2$e per km). In Fig. 2a, this can be visualised by seeing the points moving further away from the diagonal line, for the more ambitious scenarios.

There are approximately 1% of cases in Fig. 2a where, in the "No scenario" approach, HEVs have a lower carbon footprint than BEVs (indicated by a few data points in the BEV area of the chart). These occur in high-carbon electricity regions, such as Eastern Europe (EEU) when vehicles have a low total mileage of around 100,000 km. In the most pessimistic "No climate action" scenario, this drops to well below 1%, and to 0% in all other future scenarios. Similarly, Fig. 2b indicates that BEVs will have a lower mean carbon footprint of 20–31% than PHEVs in all future scenarios, compared to 15% expected in typical LCA. The pLCA reveals that BEVs,

## a) Global electricity production

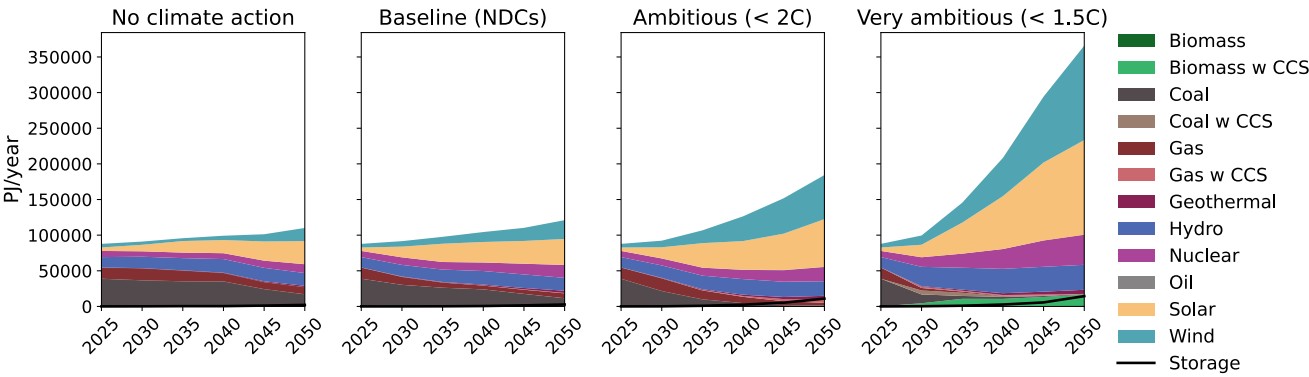

## b) Global diesel production

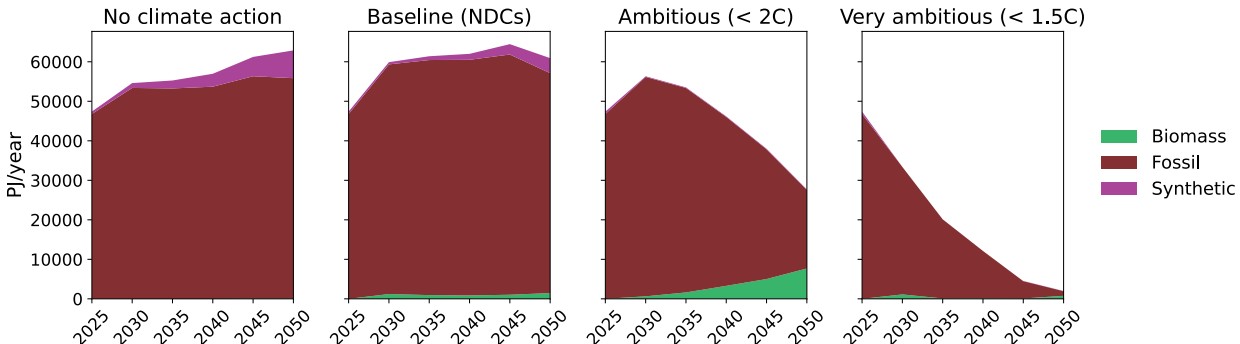

## c) Global hydrogen production

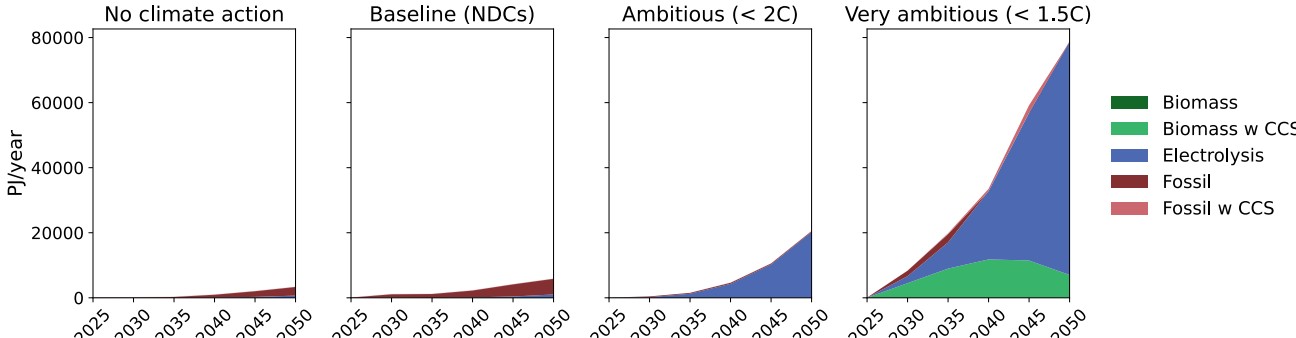

**Fig. 1 | Future global energy supply technology mixes.** The key production mixes are **a** electricity, **b** diesel, and **c** hydrogen from TIAM-UCL across all scenarios. TIAM-UCL is a global energy-economy model representing energy systems across 16 regions, aiming for cost-effective solutions to meet future energy needs while considering climate, resource, and technological constraints. Scenarios are based on the SSP2 "Middle of the Road" pathway under four climate futures of RCP6.0 - "No climate action", RCP4.5 - "Baseline (NDCs)", RCP2.6 - "Ambitious ( < 2 °C)", and RCP1.9 - "Very ambitious (1.5 °C)".

followed by PHEVs, receive the most major carbon reductions compared to HEVs. This is because, during their use phase, projected decarbonisation is greater for electricity than fuel supply.

FCEVs present a more nuanced case. In Fig. 2c, in future scenarios excluding "Very ambitious (1.5 °C)", BEVs have 14–19% lower mean carbon footprints than FCEVs and are lower in 88–96% of individual cases. This is because the hydrogen supply mainly relies on fossil reforming or energy-intensive electrolysis that is not sufficiently decarbonised by low-carbon electricity supply. However, in the "Very ambitious (1.5 °C)" scenario, the comparison flips, and FCEVs show a 70% lower mean carbon footprint than BEVs and are lower in 100% of individual cases (Fig. 2c) with the same trend for PHEVs (Fig. 2e) and HEVs (Fig. 2f). This is due to noteworthy future hydrogen production from BECCS, a carbon-negative technology, which considerably reduces the FCEV carbon footprints in this scenario only (See Fig. 1). In Fig. 2e, it is difficult to discern a clear trend for most scenarios. Still, FCEVs have a slightly lower mean carbon footprint than PHEVs in all but the "Very ambitious (1.5 °C)" scenario, where FCEVs have a clear advantage. Meanwhile, Fig. 2f indicates that FCEVs have a lower mean carbon footprint than HEVs in all future scenarios.

Table 1 summarises the mean carbon footprint of each passenger vehicle. In future scenarios, excluding the "Very ambitious (1.5 °C)" case, BEVs show the lowest mean carbon footprints for vehicles produced in 2025. FCEVs have the second-lowest carbon footprints, closely followed by PHEVs, while HEVs consistently exhibit the highest carbon footprints. For future passenger vehicles produced from 2035, BEVs retain the lowest mean carbon footprint. PHEVs are now the second-lowest in the "No climate action" and "Baseline (NDCs)" scenarios, surpassing FCEVs due to greater decarbonisation. HEVs consistently show the highest mean carbon

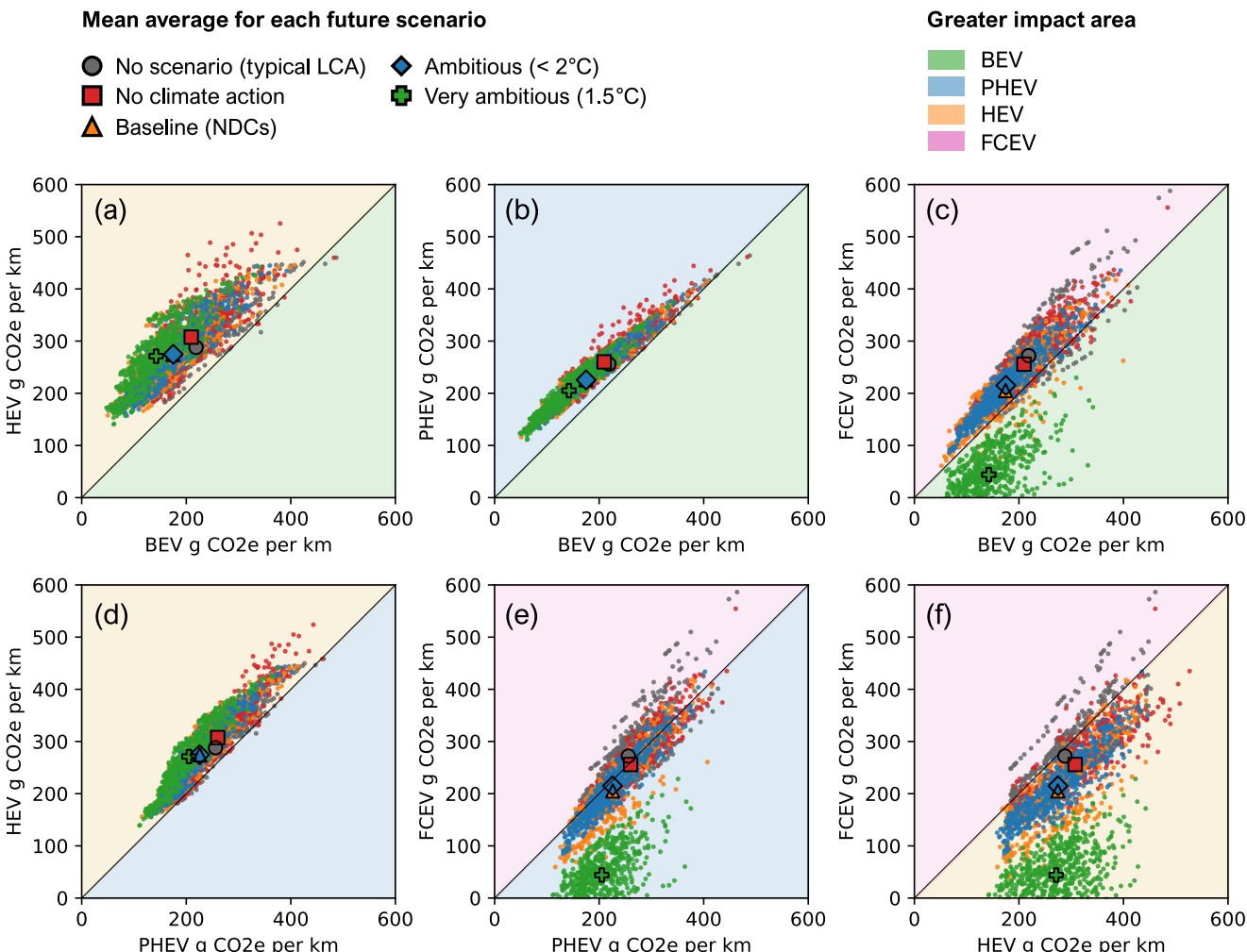

**Fig. 2 | Comparative Monte Carlo results for the life-cycle carbon footprint of passenger vehicle powertrains for four future scenarios.** The production year is 2025 to show present-day vehicles that will be used and reach their end-of-life in 2035–2045. Results represent 5000 unique configurations computed equivalently for BEV, PHEV, HEV, and FCEV. The mean points of each scenario are displayed in their respective colour and shape, defined in the legend as "Mean average for each future scenario". For example, **a** represents comparable pairwise results for a BEV and HEV. Each point reflects a BEV vs. HEV result from the equivalent configuration of the same mileage, region, lifetime, future scenario, size etc. If a point is in the HEV area of the chart (shaded light yellow, represented in the "Greater impact area" legend), then the HEV had a higher carbon footprint than the BEV in that configuration. The opposite is true if a point is in the BEV area of the chart (shaded light green). **b, c** show the comparison for BEV vs. PHEV and FCEV, respectively; **d, e** compare PHEV vs. HEV and FCEV, respectively; and **f** compares HEV vs. FCEV.

footprints with increasing margins due to fewer future decarbonisation benefits. As previously discussed, the "Very ambitious (1.5 °C)" scenario for current and future vehicles maintains that FCEVs have the lowest mean carbon footprint, subject to the high sensitivity of BECCS uptake.

**Contribution & decarbonisation analysis.** Figure 3 provides an in-depth case study and comparative contribution analysis of passenger vehicles in CHI, which represents the case with the smallest carbon footprint margins. It helps explain and analyse the results in Fig. 2 and Table 1, highlighting key differences between typical LCA and pLCA approaches, carbon hotspots, and future decarbonisation potential for each vehicle.

Figure 3a shows that for vehicles produced in 2025, a typical LCA without future scenarios finds that all vehicles have relatively similar carbon footprints over 200,000 km. The BEV, PHEV, and FCEV exhibit greater production impacts than the HEV due to increased component demands, such as lithium-ion batteries and fuel cells. However, the electricity and hydrogen supply for the BEV, PHEV, and FCEV have lower carbon footprints than the diesel supply and exhaust emissions in the HEV. The BEV and PHEV could also have additional benefits outside the system

boundaries from future battery recycling (shown as negative bar "credits") such as the recovery of nickel sulphate that has high embodied carbon.

Figure 3b illustrates why the differences among the vehicles become pronounced in the pLCA approach (in the case of the "Ambitious ( < 2 °C)" future scenario). For example, the BEV carbon footprint reduces by 31% from Fig. 3a, b since the electricity supply in CHI is expected to decarbonise over the 15-year vehicle lifetime. Meanwhile, the carbon footprint of other vehicles is reduced by a smaller 3–19%. As a result, the BEV carbon footprint stands out as the lowest in the pLCA approach (Fig. 3b) compared to what the typical approach (Fig. 3b) would find. Figure 4a further shows how the trends in Fig. 3 generalises to all regions finding that the pLCA approach (blue bars), using the "Ambitious ( < 2 °C)" future scenario, consistently amplifies the advantage of BEVs over HEVs, compared to the typical LCA approach (maroon bars).

Figure 4b–d illustrate the underlying reasons for this by showing the regional carbon footprints for the supply of electricity, diesel, and hydrogen, comparing the typical LCA and pLCA approaches. Figure 4b shows that the average carbon footprint of the 2025–2040 electricity supply used in the pLCA is much lower than the static 2025 supply assumed in the typical LCA due to the projected decarbonisation of regional electricity mixes. Since

electricity supply is a major hotspot, decarbonisation substantially lowers the carbon footprint of BEVs in the pLCA compared to the typical LCA.

Similarly, Fig. 4c illustrates that diesel does also decarbonise mainly via biodiesel uptake in the pLCA approach, as seen in the CHI case by 41%. However, Fig. 3a reveals that diesel supply represents a relatively minor contribution to the HEV carbon footprint since it is outweighed by the dominant exhaust emissions. Therefore, even when diesel decarbonisation is accounted for across 2025–2040, Fig. 3b shows that HEV carbon footprint reduction is a relatively small 11%. Meanwhile, Fig. 4d indicates that hydrogen production in CHI achieves minimal decarbonisation initially, as fossil-based chemical reforming remains prevalent in earlier years. Considerable hydrogen decarbonisation occurs later with the adoption of electrolysis and renewable electricity, although this happens sooner in regions, such as WEU. Therefore, the FCEV carbon footprint reduces minimally from the typical LCA in Fig. 3a to the pLCA in Fig. 3b.

Figure 3c additionally captures the potential decarbonisation of production with a pLCA of vehicles produced in 2035, used between 2035 and 2050, and reaching their end-of-life in 2050. From 2025 to 2035, production carbon footprints decrease by 21–25%. For example, the carbon footprint of manufacturing energy storage components (e.g., lithium-ion batteries and fuel cells) and gliders (e.g., vehicle body) reduces due to upstream dec-

arbonisation of electricity and fuel mixes, and uptake of electric and secondary steel production. Continued decarbonisation across 2035–2050 achieves further carbon footprint reductions in the vehicle use phases, where Fig. 3b, c, hydrogen supply for FCEVs achieves substantial reductions while diesel and electricity hotspots become diminished for HEVs, PHEVs, and BEVs. Furthermore, all powertrains produced in 2035 benefit from efficiency improvements based on inventory projections, with PHEVs showing the most considerable improvements, leading to sizable reductions in exhaust emissions. With future decarbonisation shown for vehicles produced from 2035 in Fig. 3c, the carbon footprint outcomes remain consistent with Fig. 3b: BEVs continue to show the lowest carbon footprints, followed by PHEVs and FCEVs. HEVs display a higher carbon footprint by a larger margin due to their unmitigated exhaust emissions.

Overall, the pLCA approach indicates that when future scenarios are considered in Fig. 3b, c, all passenger vehicles are forecasted to have lower carbon footprints than the typical LCA approach, as in Fig. 3a, leading to the multi-case results and trends seen in Fig. 2. However, BEVs are projected to see the most meaningful reductions due to the scale and magnitude of electricity decarbonisation, greater than diesel or hydrogen. As a result, BEVs are projected to have the lowest carbon footprints in most cases.

### Impact of regional electricity carbon intensities on breakeven mileages

Figure 5 further evaluates the influence of the pLCA approach by determining the minimum mileage a BEV must be driven to achieve a lower carbon footprint than an HEV, considering various regional electricity and diesel carbon intensities and future scenarios. Each data point represents a region and future scenario combination, with BEV vs. HEV results assessed across different mileages for a Medium SUV (chosen due to the BEV vs. HEV having the smallest carbon footprint margins for this vehicle size). Each data point indicates the mileage (x-axis) at which a BEV, despite its initially higher production impacts, achieves a lower life-cycle carbon footprint than an HEV, given the average regional electricity carbon intensity (y-axis).

Figure 5 shows that most BEV breakeven mileages across different regions and future scenarios are below 100,000 km. Breakeven mileages exceeding 100,000 km mainly appear in the "No scenario" data points, which assume higher average carbon intensities represented in the typical LCA approach. When the pLCA approach considers future scenarios, most breakeven mileages are considerably lower due to projected reductions in electricity carbon intensities. However, there is no smooth trend since varying diesel supply carbon intensities also influence these outcomes. Few pLCA data points exceed 100,000 km, which are the "No climate action" scenario in EEU and South Korea, and the "Baseline (NDCs)" scenario in

**Table 1 | Summary of comparative MC results for the mean carbon footprints of passenger vehicles**

| Scenario | Production year | BEV | PHEV | FCEV | HEV |
|---|---|---|---|---|---|
| No scenario | 2025 | 0% | +15% | +21% | +24% |
| No climate action | 2025 | 0% | +20% | +18% | +32% |
| | 2035 | 0% | +12% | +13% | +41% |
| Baseline (NDCs) | 2025 | 0% | +22% | +14% | +36% |
| | 2035 | 0% | +15% | +16% | +48% |
| Ambitious (< 2 °C) | 2025 | 0% | +23% | +19% | +37% |
| | 2035 | 0% | +15% | +6% | +49% |
| Very ambitious (1.5 °C) | 2025 | 0% | +31% | -70% | +47% |
| | 2035 | 0% | +29% | −103% | +66% |

For each unique scenario and production year, the BEV mean carbon footprint is set as the reference point at 0%. The mean carbon footprints of other vehicle types are then expressed as the percentage by which they exceed this reference point. For example, in the first row under the "No scenario" with a 2025 production year, the BEV mean carbon footprint is set at 0%. The PHEV, FCEV, and HEV have average carbon footprints that are 15, 21 and 24% higher than the BEV, respectively.

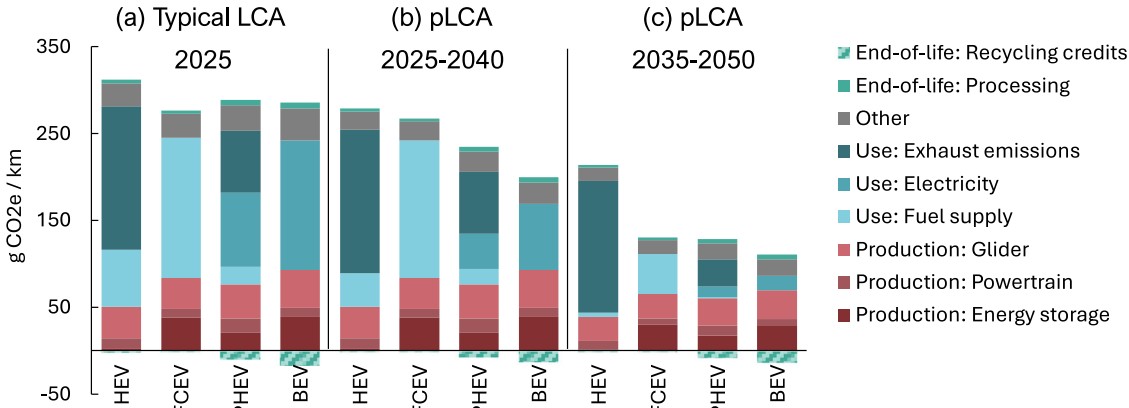

**Fig. 3 | Contribution analysis of current and future passenger vehicles.** The figure shows **a** typical LCA for a production year of 2025, and time-adjusted pLCA for production years of **b** 2025 and **c** 2035. The selected case study is the China region, Medium SUV size, "Ambitious (< 2 °C)" climate mitigation scenario, 200,000 km mileage, 15-year lifetime, and hydrometallurgical battery recycling. Potential battery recycling credits outside the system boundary are shown for reference. However, these credits are not used to "reduce" the carbon footprint of vehicles, such as the results in Fig. 2.

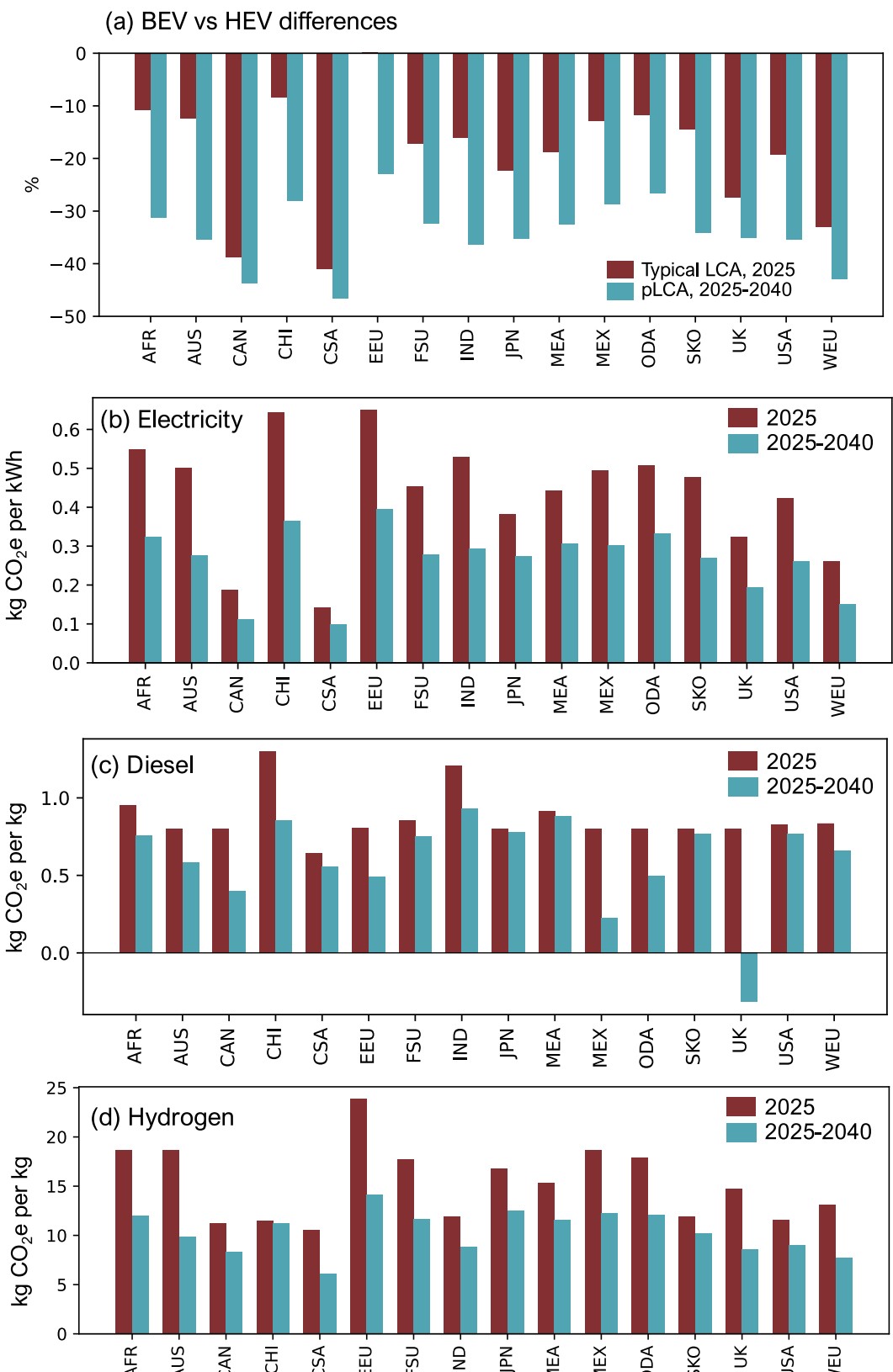

**Fig. 4 | Carbon footprint differences of vehicles, electricity, diesel, and hydrogen.** **a** The regional carbon footprint percentage difference between BEV and HEV in different regions using typical LCA and pLCA approaches with the "Ambitious (< 2 °C)" future scenario. This assumes Medium SUV vehicles have a 15-year lifetime and 200,000 km total mileage. Note that the typical EEU case has a near 0% difference and is not a missing data point. **b–d** show the regional electricity, diesel and hydrogen production carbon footprints for 2025 and the average carbon footprint between 2025 and 2040 in the "Ambitious (< 2 °C)" future scenario.

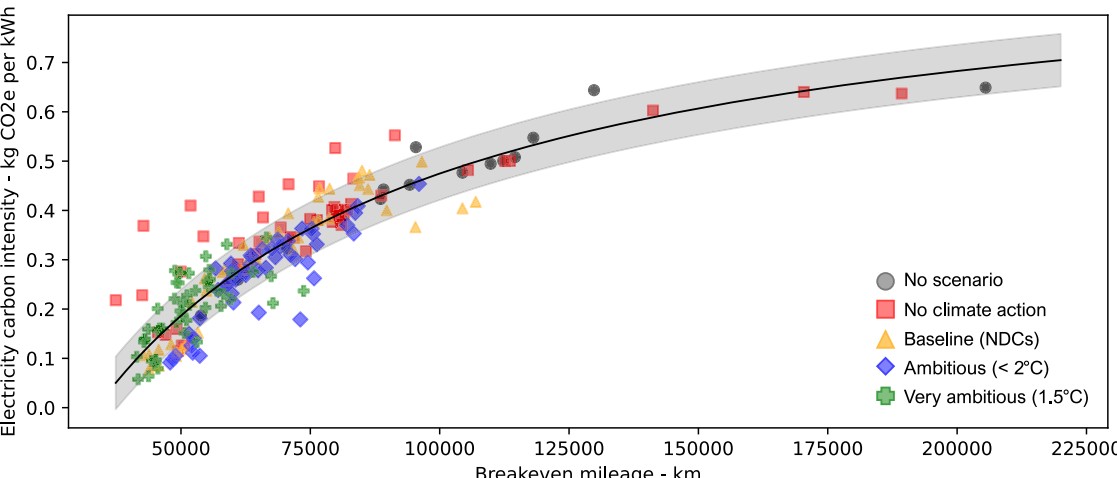

**Fig. 5 | The graph illustrates the minimum mileage (x-axis) a BEV must be driven to achieve a lower life-cycle carbon footprint than a HEV, given an average carbon footprint for electricity (y-axis).** The analysis includes all 16 regions, four future scenarios, and the "No scenario" option, considering vehicle lifetimes of 10, 15, and 20 years. Results were calculated across various mileages for BEV and HEV models produced in 2025. The breakeven mileages could be determined given the scores over different mileages and the average electricity carbon intensity for each region, scenario, and lifetime. Each point represents a unique region, scenario, and lifetime combination. The Medium SUV size was chosen as it represents the case where the differences between BEV and HEV are the smallest, while other vehicle sizes tend to show larger differences. Other vehicle sizes would result in lower breakeven mileage with lower carbon intensities. The scatter in the data points is due to variations in the average carbon intensity of electricity and the diesel supply for the HEV. For instance, in regions with similar average carbon intensity, a lower carbon footprint from diesel supply will result in the BEV needing to be driven further to reach the breakeven point with the HEV since the HEV.

Mexico. This aligns with the <1% cases where HEVs outperformed BEVs in Fig. 2a. On average, BEV breakeven mileages are 87,000 km (0.381 kg CO2e/kWh) for "No scenario", 70,000 km (0.343 kg CO2e/kWh) for "No climate action", 64,000 km (0.278 kg CO2e/kWh) for "Baseline (NDCs)", 63,000 km (0.258 kg CO2e/kWh) for "Ambitious ( < 2 °C)", and 51,000 km (0.191 kg CO2e/kWh) for "Very ambitious (1.5 °C)". Future scenarios reducing the needed breakeven mileages increases confidence that BEVs can payback their production emissions sooner. In contrast, higher needed breakeven mileages naturally increase uncertainty about the timing and reliability of their overall carbon benefits. However, more granular regions and provinces not represented in TIAM-UCL, with electricity carbon intensities above 0.600 kg CO2e/kWh, may show little to no carbon footprint advantage for BEVs over HEVs.

### Global sensitivity analysis

Figure 6 presents the GSA results, highlighting the pLCA inputs with the greatest impact on passenger vehicle carbon footprints. Figure 6a (left-hand column) shows that vehicle size substantially impacts results across all vehicles since it directly determines manufacturing and use-phase demands (except for FCEVs). For instance, larger BEVs require bigger batteries, thereby increasing the impacts associated with manufacturing. Also, heavier vehicles decrease their energy efficiency, consuming more electricity or fuel per kilometre. Total mileage also strongly influences results by affecting lifetime electricity, diesel, hydrogen consumption and exhaust emissions. The region and future scenario, which affect electricity and fuel carbon intensities, become increasingly important for HEVs, followed by PHEVs and BEVs. For FCEVs, the future scenario is the most considerable factor due to the "Very ambitious (1.5 °C)" case.

Figure 6b shows the pLCA inputs that most influence carbon footprint differences between BEVs, PHEVs, and FCEVs, with HEVs as the reference due to their consistently highest carbon footprints. Size is less critical here, as production and use-phase changes similarly affect all vehicles. Instead, region and future scenario become most influential due to substantially varying electricity and hydrogen supply carbon intensities, majorly impacting BEVs, PHEVs, and FCEVs relative to HEVs. While regional diesel production does decarbonise, the effect on HEVs is smaller, making changes in electricity and hydrogen supply the key drivers of carbon footprint differences.

Figure 6c, d build on this to show these differences within specific regions. In EEU, with its high-carbon electricity in 2025, the future scenario is the most influential for carbon footprint differences in BEVs and PHEVs due to the substantial future decarbonisation potential of the electricity mix. In contrast, the future scenario is not as important in Central and South America, where the electricity grid is already low-carbon. However, for FCEVs in both regions, the future scenario remains dominant, particularly due to the impact of the "Very Ambitious (1.5 °C)" scenario.

The overall GSA demonstrates that size, total driven mileage, and regions are key determinants of absolute carbon footprints. However, the combined effects of region and future scenario are pivotal in determining the comparative carbon footprints, especially for regions with the greatest future decarbonisation potential. Meanwhile, inputs like lifetime and battery recycling generally have relatively minimal influence, although they could become more important when considering other impact categories, such as metal depletion.

### Discussion

We show that incorporating future decarbonisation scenarios is crucial for determining comparative carbon footprints. All current and future passenger vehicles are projected to have lower carbon footprints due to the progressive decarbonisation of electricity, fuels, hydrogen, and more. This reduction, along with other factors, is often not captured in typical LCA approaches based on historical data. Compared to previous pLCA, this study enhances rigour through several advancements. First, it considers broader energy system transitions, not just electricity[24,25] but also the roles of fuels and hydrogen in decarbonisation pathways important for non-electric powertrains, while also encompassing future manufacturing impacts by accounting for transitions in sectors such as steel. Second, it expands the variety of future scenarios, covering four 1.5–3.0 °C decarbonisation pathways, whereas previous studies typically considered only two scenarios[6,25]. Third, it introduces a new TIAM-UCL model, exploring outcomes beyond just REMIND[6] and IMAGE[24,25]. Additionally, it improves variations in the pLCA model by refining how life-cycle stages are time-adjusted and ensuring that both the LCA background and foreground systems consistently represent the same IAM future[6]. Lastly, the study conducts a comprehensive comparative analysis using a MC simulation of 5000 unique

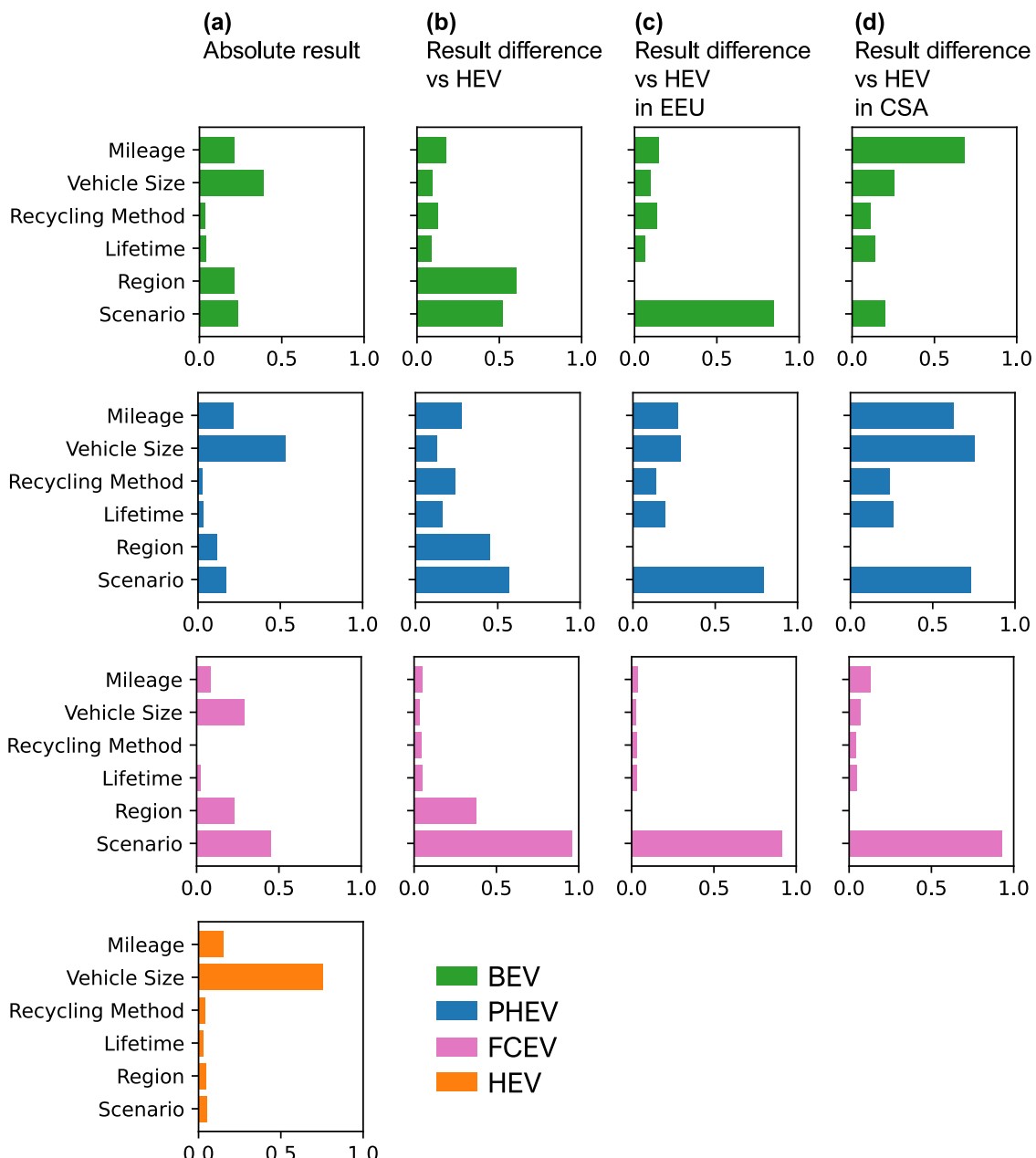

**Fig. 6 | Sobol total order indices from global sensitivity analysis for time-adjusted parameters of BEV, PHEV, and FCEV. a** single vehicle absolute carbon footprints scores **b** percentage difference in results between vehicle and HEV **c** percentage difference in results between vehicles and an HEV in China—a high-carbon electricity region. **d** percentage difference in results between vehicles and an HEV in Central and South America (CSA)—a low-carbon electricity region.

configurations and quantifies the effects of contextual factors such future scenario choice using GSA.

BEVs consistently show the lowest carbon footprints among passenger vehicles across different sizes, mileages, regions, and future scenarios, in cases where they are driven over 100,000 km. In less than 1% of cases in most scenarios, other vehicles can match BEV results, but only in scenarios involving the combined effects of larger vehicles, pessimistic decarbonisation futures, low total mileage, and regions with high-carbon electricity grids. HEVs consistently have the highest carbon footprints, as future decarbonisation of fuels like diesel, including biomass-derived options, remains insufficient in all projections. In contrast, even in pessimistic climate scenarios, electricity decarbonisation will continue to benefit BEVs and PHEVs. Similarly, despite current fossil-based hydrogen, FCEVs can outperform HEVs, with more optimistic projections showing potential for hydrogen decarbonisation.

While these findings generally align with other LCA and pLCA studies, some studies have suggested that BEVs may have higher carbon footprints than combustion vehicles, such as HEVs and ICEVs in regions with high-carbon electricity grids like CHI[6,9,31] or India[6,8]. In contrast, our study finds that current BEVs can retain the lowest carbon footprints even in these regions. The key distinction in our study is that by consistently time-adjusting the pLCA with future decarbonisation trajectories, we project a lower average carbon intensity for BEV electricity supply than typical LCA. This is especially true in high-carbon regions like CHI, which can be expected to considerably decarbonise over the next 10–20 years in all future trajectories. While diesel supply could also decarbonise, in these scenarios, it does not sufficiently become carbon-negative to compensate for the dominant exhaust emissions in HEVs.

These conclusions focus on representing global and major regional future trends, but overgeneralisation to specific regions is cautioned. A

key limitation of the consistent application of IAM scenarios to pLCA is the loss of regional resolution due to the aggregation of multiple regions, as seen in groupings, such as WEU or EEU. For instance, this aggregation may overlook edge cases in energy mix carbon intensities, such as the exceptionally high and low electricity mixes of Estonia and Norway, respectively[32]. Furthermore, while major regions, such as the United States, CHI, and India do have their independent indicators in TIAM-UCL, considerable provincial variations in energy mixes still introduce uncertainty[33]. The trend analysis on regional electricity carbon intensity illustrates how this factor impacts BEV breakeven mileage, accounting for potential variations that are not explicitly represented in specific TIAM-UCL regions. In regions with greater carbon-intensity electricity, BEVs must cover considerably more miles to compensate for their higher production emissions and achieve parity with HEVs. This adds uncertainty to their ability to achieve a meaningfully lower carbon footprint. However, even in the "No climate action" scenario, accounting for future decarbonisation could reduce this breakeven mileage, strengthening confidence in BEVs achieving meaningful carbon footprint reductions. On the other hand, TIAM-UCL is limited by its yearly temporal resolution, preventing it from capturing in-year variations in energy mixes, such as the hourly intermittency of renewables for charging[34]. Therefore, applying these broader conclusions to specific contexts would ultimately need an enhanced geographical and temporal resolution of prospective energy mixes that cannot be captured by IAMs.

Other vehicle types can potentially give lower impacts than BEVs in specific cases. In the "Very ambitious (1.5 °C)" scenario, FCEVs can achieve the lowest carbon footprints. However, this relies on the accelerated deployment of carbon-negative hydrogen production from BECCS, which is unrealistic due to the current low technological readiness[35]. In the short term, hydrogen supply decarbonisation from electrolysis is a more likely case, as seen in the "Baseline (NDCs)" and "Ambitious (< 2 °C)" scenarios, where FCEVs typically show higher carbon footprints than BEVs, but similar to PHEVs and lower than HEVs. PHEVs can be an intermediate option between BEVs and HEVs, with the pLCA study assuming that about 47% of PHEV mileage is driven in electric mode[36]. However, this is sensitive to driver behaviour, where greater utilisation of electric mode could lead to lower carbon footprints than BEVs and lower utilisation could result in greater carbon footprints than HEVs. Since drivers can often underutilise PHEV electric mode[37], PHEVs are a high-risk option with greater carbon footprint uncertainty.

In summary, our study supports BEVs as the most reliable climate mitigation option for passenger cars, consistently showing the lowest carbon footprints with expected electricity supply decarbonisation to meet climate objectives. This conclusion complements and reinforces previous pLCA study findings that used IMAGE[24,25] and REMIND[6] scenarios by methodological improvements and expanding to TIAM-UCL to enhance rigour. Continued strong electricity decarbonisation is incentivised to maximise BEV advantages and minimise their carbon footprints. However, BEVs have the highest manufacturing carbon footprints, with their life-cycle benefits only realised later in the use phase. To further minimise carbon footprints, strategic BEV adoption is essential. Manufacturers should prioritise energy reduction, sustainable materials, manufacturing efficiency, and smaller vehicle sizes to reduce material demands and improve energy efficiency, as production impacts remain major even with future improvements.

Although the study focused on advancing the prospective methodology and assessing the impact of variation of prospective indicators, further work could explore the uncertainty in manufacturing impacts arising from variations in battery chemistry and raw material sourcing, such as lithium, nickel, and cobalt[38]. In addition to improvements in refining the geographic and temporal resolutions, future work could expand to assess environmental impacts beyond carbon footprints, including acidification, eutrophication, and resource depletion, for a more holistic evaluation[26]. Nonetheless, for long-lived technologies, such as passenger vehicles, but also in other sectors such as construction and energy, incorporating future scenarios in time-adjusted pLCA is essential. Considering potential

decarbonisation trajectories provides more representative results and can lead to distinctly different outcomes, enabling more reliable carbon footprint assessments. This approach helps avoid misleading decision-making based on present-day impacts that may not account for the uncertainties of future developments.

## Methods

### Overview

The system boundaries followed a "cradle-to-grave" scope, encompassing raw materials, manufacturing, use, and end-of-life stages. The scope covered the following types of passenger vehicles: BEV, PHEV, and HEV, with decreasing degrees of electrification, respectively. Since HEVs are combustion-based powertrains with very small batteries (< 5 kWh) that help increase fuel economy from regenerative braking, they are used as the dominant combustion powertrain instead of pure ICEVs. Although hydrogen FCEVs have limited market shares due to early maturity and lacking hydrogen infrastructure, they are also included due to potential future interests[39]. The study's temporal scope spanned from 2025 to 2050 in 5-year steps, and its geographical scope encompassed 16 world regions as defined in TIAM-UCL.

Passenger vehicle results were compared based on the functional unit of "per 1 km driven over the lifetime". The LCA followed an attributional system model. For the overall LCA results, the cut-off end-of-life approach was applied, meaning that potential credits from recycled products were not included to avoid possible double counting from existing recycled content[40]. However, to represent potential future benefits from battery recycling outside the system boundary[14], credits were presented separately in a contribution analysis following the EN 15804 approach[41] applied in the construction sector. For general interest, credits were included in the GSA to investigate their potential influence on LCA results if included.

Foreground systems were generated, containing the primary cradle-to-grave inventories of various passenger vehicles (under Foreground system). Meanwhile, several background systems containing electricity, diesel, and hydrogen supply mixes were produced based on future scenarios and years (under Background system). Corresponding foreground and background systems were matched using Brightway2[42] to form the final inventories for conducting pLCA and further analysis.

### Foreground system

The "Carculator" LCA model Python package was used as the basis of passenger vehicle production and use inventories[6]. Carculator is a parameterised model that generates life-cycle inventories based on various vehicle configurations derived from European statistics and calibrated using data from over 15,000 vehicles[25,43,44]. This provided a consistent and reproducible inventory methodology for passenger vehicles.

Four powertrains, BEV, PHEV, HEV, and FCEV, with diesel-based engines assumed for PHEV and HEV, were selected across seven vehicle sizes: mini, small, lower medium, medium, large, medium SUV, and large SUV. Diesel was chosen as the primary combustion fuel, rather than gasoline, due to its better technology representation in the TIAM-UCL future scenarios. For implications, diesel tends to have a lower carbon footprint compared to gasoline in the Carculator model, making it a more optimistic assumption for combustion powertrains than gasoline. Additionally, 2020, 2030, 2040, and 2050 vehicle models are selected to represent expected future improvements in vehicle efficiencies and emission standards (e.g., EURO-7). Detailed modelling and assumptions can be found in the Carculator documentation, but key assumptions are described in the next paragraphs.

Vehicles of the same size class are designed to follow the WLTP driving cycle with the same speed profile, seating capacity, and roughly equivalent engine power (determined by the driving cycle) to ensure fair powertrain comparisons. Vehicle energy consumption and tailpipe emissions were calibrated against the standardised WLTP driving cycle, with electricity consumptions ranging from 19.2–28.5 and 11.8–21.4 kWh per 100 km for BEV and PHEVs respectively, and 3.7–7.5 and 2.0–3.0 L per 100 km for

https://doi.org/10.1038/s43247-025-02447-2                                                                                          **Article**

HEV and PHEV, respectively, with PHEVs assuming a 47% electric utility factor (i.e., share of kilometres driven in battery-depleting mode). The NMC-811 lithium-ion battery chemistry was assumed for all relevant vehicles with pack sizes of 14–80 kWh since high nickel batteries are the most prevalent chemistry[45]. The fuel cell system for FCEVs was based on the proton exchange membranes technology.

These inventories were combined with three battery recycling inventories based on pyrometallurgical, inorganic hydrometallurgical, and direct recycling processes[14]. The foreground inventories underwent further data processing, such as regionalising electricity, hydrogen, and diesel supplies for vehicles used in different regions and modifying inventory structure for ease of time-adjustment, e.g., clearly separating production, use, and end-of-life activities. Finally, foreground inventory copies were generated to match each corresponding TIAM-UCL scenario and year of the background system.

## Background system
TIAM-UCL is a global energy-economy model representing energy systems across 16 regions from primary sources to end-use, aiming for cost-effective solutions to meet evolving energy needs under different climate, natural resource, and technological constraints[46]. Preceding work with Premise mapped over 200 technology variables across 16 global regions to over 300 LCA processes representing future technological changes across seven major sectors, including electricity, diesel, hydrogen, and other products such as steel[26]. Scenario files from TIAM-UCL were fed into Premise and ecoinvent v3.9.1 to generate background prospective life cycle inventories (pLCIs). The TIAM-UCL scenarios represented the SSP2 "Middle of the Road" future pathway that assumes moderate climate change mitigation and adaptation[47] challenges. This was constrained under four different climate futures, as represented by representative concentration pathway (RCP) scenarios limiting radiative forcings by the year 2100 as follows (where RCP6.0, for example, refers to 6.0 watts per square metre (W/m²)).

- SSP2-RCP6.0—"No climate action". A gradual pathway for reducing greenhouse gas emissions, aiming to limit global warming to between 2.6 and 4.8 °C by 2100. While some measures are implemented to curb emissions, they fall significantly short of the ambitious targets set by the Paris Agreement. The transition away from fossil fuels and the adoption of renewable technologies proceed at a sluggish pace.
- SSP2-RCP4.5—"Baseline, NDCs". The current path to reducing greenhouse gas emissions aims to limit global warming to between 2.0 and 2.5 °C by 2100. This approach involves more substantial efforts to curb emissions than the RCP6.0 scenario, leading to an earlier peak and gradual decline in GHG emissions. However, it still falls short of fully meeting the Paris Agreement's recommendations.
- SSP2-RCP2.6 – "Ambitious ( < 2.0 °C)". A swift greenhouse gas reduction pathway aligned with the Paris Agreement's objectives to prevent severe climate change impacts, aiming to limit global warming to well below 2.0 °C by 2100. This approach necessitates extensive decarbonisation of the global economy, a major transition to renewable energy sources, and broad implementation of carbon capture and storage technologies.
- SSP2-RCP1.9 – "Very ambitious (1.5 °C)". The most accelerated greenhouse gas reduction pathway aligned with the Paris Agreement aims to limit global warming to 1.5 °C by 2100. This scenario involves even more aggressive decarbonisation and carbon removal efforts than RCP2.6, including the swift phase-out of fossil fuels and the large-scale deployment of carbon dioxide removal technologies.

For each scenario, TIAM-UCL output variables related to production volumes, consumption, carbon capture, and efficiencies for various energy technologies across electricity, steel, fuels, cement, biomass, heat, and direct air capture were reported[48]. These were provided across 16 world regions between the years 2005 to 2100. However, this study's temporal focus is from 2025 to reflect the present day to the year 2050 due to increasing model uncertainty in the later years. Premise v2.1.1 and ecoinvent v3.9.1 were used

to generate background pLCI corresponding to TIAM-UCL sector transitions, future scenarios, and years matched with the foreground inventories.

## Time-adjustments
The impacts of foreground and background systems were time-adjusted and distributed across different periods of the vehicle's lifetime, considering eight key model inputs. For example, the vehicle's production, use, and end-of-life inventories were characterised by inputs such as powertrain, size, production year, region, and recycling method. Inputs like production year, lifetime, mileage, and future scenario were used to time-adjust and distribute vehicle impacts in 5-year increments.

For instance, the production year would define "year 0" of the vehicle when all manufacturing and upstream impacts occur. The vehicle's lifetime (e.g., 15 years) and mileage (e.g., 210,000 km) would set the use phase period (e.g., years 0–15) and the end-of-life point (e.g., year 15). Use-phase impacts would be calculated as the average between year periods, such as 2025–2030, 2030–2035, and 2035–2040, assuming equal mileage distribution (e.g., 70,000 km across each period). The chosen future scenario (e.g., "Ambitious ( < 2 °C)") determines the specific foreground and background systems from which the vehicle inventories and impacts are drawn, reflecting future energy system developments based on TIAM-UCL projections. Impacts were calculated as cumulative totals and converted to a per 1 km FU basis.

## Life cycle impact assessment
The life cycle impact assessment (LCIA) method applied was the IPCC 2021 100-year GWP[27]. We adapted this GWP method by assigning a characterisation factor of $-1/+1$ to the uptake and emission of non-fossil $CO_2$, allowing for the inclusion of net negative emission technologies in TIAM-UCL. For example, atmospheric $CO_2$ absorbed by biomass and stored during electricity generation with CCS results in net carbon removal despite associated parasitic emissions.

## Comparative LCA
To account for a wide range of vehicle configurations, use cases, and future scenarios, we employed a MC approach. Multiple iterations were sampled and computed in the pLCA, considering combinations of size (7), region (16), mileage (100,000–300,000 km), lifetime (10–20 years), and scenario (4). A dependent sampling method using comparative MC[49,50] was adopted to ensure valid powertrain comparisons. This approach ensured that BEV, PHEV, HEV, and FCEV comparisons were based on equivalent inputs within each MC simulation. For instance, a BEV vs. HEV comparison used the same mileage (e.g., 200,000 km) among other inputs. Given the computational intensity of running all possible configurations, we ran 5000 comparative MC samples, which converged on a representative view of the variation in the comparisons across configurations. These samples focused solely on input variations and excluded uncertainties from foreground inventories or background databases.

## Global Sensitivity Analysis
GSA was conducted to identify which pLCA inputs caused the most variation in the results. We used Sobol's indices with the Saltelli sampling method[51], a variance-based approach that attributes output variance to individual input variables and their interactions. This method effectively handles non-linear and non-monotonic relationships and is widely used for its ability to manage complex interactions and provide robust sensitivity insights. The SALib Python library was used for this analysis[52]. Both first-order and total-order Sobol indices were considered to capture the independent contributions of variables and their interactions. Categorical variables were modelled as uniform distributions rounded to integer values. For instance, the "Future scenario" variable (No Scenario = 0, No climate action = 1, Baseline (NDCs) = 2, etc.) was sampled between −0.5 and 4.5. GSA results focused on comparison outcomes (e.g., the influence of parameters on the percentage difference between BEV and HEV) rather than absolute outcomes (e.g., the impact of parameters on BEV alone).

**Article**

## Data availability

The authors declare that all underlying data are available within the article and Supplementary Data 1.

## Code availability

The authors declare that the full code is available in the Supplementary Data 1.

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

## Acknowledgements

J.S. reports financial support was provided by EPSRC Centre for Doctoral Training in Advanced Automotive Propulsion Systems, Grant Number: EP/S023364/1. Joris Simaitis and Stephen Allen report financial support from the UK Energy Research Centre Phase 4 research programme (EPSRC), Grant Number: EP/S029575/1. Isabela Butnar received funding through UK Research and Innovation funded CO2RE Hub, grant number NE/V013106/1. Romain Sacchi received funding through the PRISMA project from the Swiss State Secretariat for Education, Research and Innovation (SERI) and from the European Union's Horizon Europe research and innovation programme under grant agreement No. 101081604.

## Author contributions

The contributions of each author are declared following CRediT. J.Š. contributed to conceptualisation, data curation, formal analysis, investigation, methodology, project administration, visualisation, validation, and writing - original draft. R.L. contributed to conceptualisation, methodology, visualisation, supervision, and writing, review and editing. C.V. contributed to conceptualisation, methodology, supervision, and writing, review and editing. I.B. contributed to data curation, methodology, resources, and writing, review and editing. R.S. contributed to data curation, methodology, resources, software, and writing, review and editing. S.A. contributed to conceptualisation, methodology, visualisation, funding acquisition, project administration, supervision, and writing, review and editing.

## Competing interests

The authors declare no competing interests.
