## [Transparent Peer Review file · Communications Earth & Environment]

Battery electric vehicles show the lowest carbon footprints among passenger cars across 1.5-3.0°C energy decarbonisation pathways

Corresponding Author: Dr Joris Šimaitis

Version 0:

Decision Letter:

Dear Mr Simaitis,

First of all, please allow me to apologise for the delay in sending a decision on your manuscript titled "Future carbon footprints of passenger cars". It has now been seen by 3 reviewers, whose comments are appended below. You will see that they find your work of some potential interest. However, they have raised quite substantial concerns that must be addressed. In light of these comments, we cannot accept the manuscript for publication but would be interested in considering a revised version that fully addresses these serious concerns.

We hope you will find the reviewers' comments useful as you decide how to proceed. Should additional work allow you to

- address these criticisms (that is, either to incorporate the suggestions or provide a compelling argument why the point made by the reviewer is not valid or relevant to the editorial threshold as outlined below)

AND

- meet our editorial thresholds as outlined below,

then we would be happy to look at a revised manuscript.

In the following, we list our minimum requirements for publication.

a) provide novel and fully supported insight into the impact of changes in the electricity generation mix on carbon dioxide emissions from battery electric vehicles.

b) clarify and justify your methodological choices, including scenario assumptions and combinations of vehicle types, improve regional granularity, and consider accounting for raw material sourcing; and

c) discuss the comparison between different vehicle technologies and the charging behavior of consumers and demonstrate that your data and analysis fully support all your claims.

If you choose to take up this option, please either highlight all changes in the manuscript text file or provide a list of the changes to the manuscript with your responses to the reviewers.

When resubmitting, please provide a point-by-point response to the reviewers' comments. Please submit your responses as a separate file, distinct from your cover letter where you can add responses to the Editors' comments that you do not want to be made available to the reviewers. Word files are preferred. We recommend that any figures, tables or graphs that are included in the response to reviewers are also included in the main article or Supplementary Information.

If the revision process takes significantly longer than three months, we will be happy to reconsider your paper at a later date, as long as nothing similar has been accepted for publication at Communications Earth & Environment or published

elsewhere in the meantime.

Please use the following link to submit your revised manuscript, point-by-point response to the reviewers' comments with a list of your changes to the manuscript text (which should be in a separate document to any cover letter), a tracked-changes version of the manuscript (as a PDF file) and any completed checklist:

Link Redacted

Please do not hesitate to contact us if you have any questions or would like to discuss the required revisions further. Thank you for the opportunity to review your work.

Best regards,

Pallav Purohit, PhD
Editorial Board Member
Communications Earth & Environment
orcid.org/0000-0002-7265-6960

Martina Grecequet, PhD
Senior Editor
Communications Earth & Environment

EDITORIAL POLICIES AND FORMAT

If you decide to resubmit your paper, please ensure that your manuscript complies with our editorial policies and complete and upload the checklist below as a Related Manuscript file type with the revised article:

Editorial Policy Policy requirements
(Download the link to your computer as a PDF.)

- Behavioural and social science
- Ecological, evolutionary & environmental sciences
- Life sciences

<https://www.nature.com/documents/nr-reporting-summary.zip>

For your information, you can find some guidance regarding format requirements summarized on the following checklist: (<https://www.nature.com/documents/commsj-phys-style-formatting-checklist-article.pdf>) and formatting guide (<https://www.nature.com/documents/commsj-phys-style-formatting-guide-accept.pdf>).

REVIEWER COMMENTS:

Reviewer #1 (Remarks to the Author):

By looking at the current and future carbon footprint of various passenger car powertrains, the authors address a highly relevant topic in the context of sustainable mobility. I recognize that there is a growing body of literature investigating the future environmental impacts of vehicles, including publications by some of the authors, many of which apply approaches similar to those used in this study. This article introduces notable methodological innovations, such as the temporal alignment of life cycle stages and the integration of scenarios from TIAM-UCL, which were previously unavailable to the prospective LCA community. However, my main concerns lie in the lack of clarity regarding the study's novel findings compared to existing works as well as certain assumptions made in the analysis. Please see my detailed comments below.

Comment 1: What is the actual novelty of the findings?

From an application perspective, I'm unsure what truly distinguishes the findings of this paper from previously published research. While I acknowledge the methodological innovations presented in this work (mentioned above), the major conclusions appear to largely reiterate insights already established in prior studies.

For example, the authors emphasize that “We show that incorporating future scenarios is crucial for determining comparative carbon footprints”. While this point is undoubtedly important, it has already been addressed in earlier works, including the seminal study by Mendoza-Beltran et al. (DOI: 10.1111/jiec.12825), Cox et al. (<https://doi.org/10.1016/j.apenergy.2020.115021>), and Sacchi et al. (<https://doi.org/10.1016/j.rser.2022.112475>).

For other findings, such as that “BEVs consistently show the lowest carbon footprints among passenger vehicles”, the authors acknowledge that they align with previous studies. As a main difference with previous works, the authors highlight that “some studies have suggested that BEVs may have higher carbon footprints than combustion-dominant vehicles such as HEVs and ICEVs in regions with high-carbon electricity grids like China [6,9,22], India [6,8], and Eastern Europe [6,23]. In contrast, our study finds that current BEVs will generally retain the lowest carbon footprints even in these regions.” This conclusion is noteworthy but not entirely convincing given the concerns raised in Comment #4 regarding the regional granularity of the analysis.

Overall, I do not intend this comment to diminish the value of the authors' work, which I find undoubtedly commendable. However, the novelty of the findings remains unclear from the perspective of a reader familiar with the literature. What specific new insights does this study offer compared to the already existing body of literature? Moreover, how do the methodological advances introduced in the paper enabled these contributions? These are the two questions that came to mind after reading the paper.

Comment 2: On the energy scenarios from the TIAM-UCL model

I highly appreciate the authors' effort in integrating energy scenarios from the TIAM-UCL model into prospective LCA and making them accessible to the community through premise. As the Results section starts with a description of these scenarios, it would be highly valuable to include a brief comparison highlighting the differences between the TIAM-UCL scenarios and those from REMIND and IMAGE, which are already available in premise. In short, what makes these scenarios more relevant, and what motivated the authors to undertake this effort instead of just using the existing REMIND/IMAGE scenarios? I believe providing this context would help readers better understand the important contribution of integrating these new scenarios.

Comment 3: Diesel and hydrogen cars without diesel and hydrogen?

By examining the energy scenarios considered, it becomes evident that in certain cases the production of diesel and hydrogen is either extremely low or absent (e.g., hydrogen in the “No climate action” scenario before 2035 and diesel in the “Very Ambitious” scenario by 2050). In these situations, I was wondering whether it is realistic to include these vehicle type–scenario–year combinations in the pairwise comparison analysis in Section 2.2. I don't know to what extent these combinations influence the statistical analysis of the results presented in Figure 2, but their consideration may introduce highly implausible scenarios into the analysis. The authors could consider conducting a more detailed assessment to ensure only consistent and realistic combinations are included (similar to the approach discussed in Langkau et al.: <https://doi.org/10.1007/s11367-023-02175-9>).

Comment 4: Potential limitations of the regional granularity

I'd like to address one of the main conclusions of the article: “some studies have suggested that BEVs may have higher carbon footprints than combustion-dominant vehicles such as HEVs and ICEVs in regions with high-carbon electricity grids like China [6,9,22], India [6,8], and Eastern Europe [6,23]. In contrast, our study finds that current BEVs will generally retain the lowest carbon footprints even in these regions.” This statement seems to be an oversimplification. As the authors acknowledge in the discussion, “Another limitation is that TIAM-UCL aggregates regions into 16 major areas.” This level of aggregation may overlook the significant variability in electricity grids at the country or even subregional level. For instance, Sacchi et al. (2022) provide future carbon footprints at the country level, which is not captured in this study. This limitation raises questions about the extent to which the regional granularity of this study (I acknowledge that this is a common issue when using IAMs scenarios in LCA) supports some of its broader conclusions.

It is well understood that no electric vehicle will be charged using something such as a “Western Europe power grid.” Heterogeneity exists within Western European countries or even regions within China or the US regarding their electricity mixes and potentially decarbonization objectives. Since regional variability is a core component of this work, I believe the authors should either refine the granularity of their analysis or adjust the discussion and ambition of their conclusions to better reflect this inherent limitation.

Comment 5: LCA inputs with greatest impacts

In Section 2.4, the authors present a global sensitivity analysis showing the LCA inputs with the greatest impact on the carbon footprints. While the inputs considered for the GSA are those used in their modelling, I believe there is much more that could affect the carbon footprint and that at least deserve a qualitative assessment. Notably, the supply of the raw materials used in lithium-ion batteries, which have a large influence on the carbon footprint of the battery. For example, a recent paper by Peiseler et al. (<https://doi.org/10.1038/s41467-024-54634-y>) suggests that material sourcing is more relevant than production location. Overall, I believe the authors should discuss the relevance of the inputs in light of the fact that their analysis do not consider the influence of all possible inputs.

Other minor comments:

1. In Figure 3, the authors present some avoided impacts attributed to recycling credits. The text suggests that these credits come from battery recycling. However, what about the recycling of the vehicle itself?

2. In the Discussion section, the authors mention the potential application of consequential LCA to account for system-wide changes resulting from a transition to alternative passenger car fleets. While the attributional vs. consequential LCA debate has long been a topic of discussion within the community, I wonder to what extent consequential LCA of entire sectors (and the concept of marginal suppliers) should be applied together with IAM scenarios. These scenarios already depict a specific situation for various sectors (often including the passenger transport sector) as a result of the simulation or optimization performed by the model. One may argue that only attributional LCA can be applied to assess the impacts at the fleet-level of the specific transportation sector as depicted by the IAM scenario. The point I'd like to make is that the authors raise an avenue for future research by suggesting application of consequential LCA alongside IAM scenarios, but they do not discuss the current state-of-the-art of this approach and the potential limitations.

Reviewer #2 (Remarks to the Author):

The authors present a prospective LCA study of four light duty vehicle (LDV) types on a well-to-wheel (WTW) and cradle-to-grave (CTG) basis using a global bottom-up systems engineering energy model. They find that accounting for the change in the electricity generation mix over time results in more strongly reduced CO₂ emissions from battery electric vehicles (BEVs) compared to the other LDV options. In doing that, they suggest that most analyses have taken a static approach that ignores changes in energy supply over time.

In making these claims, the authors don't seem to be aware of the vast body of energy modelling studies, starting after the first oil crisis at Oak Ridge National Lab, Pacific North-West National Lab (PNWL), IIASA, the IEA, the Paul Scherrer Institute, and other institutions. See publications by Jae Edmunds, Nebojsa Nakicenovic, Tom Kram, Clas-Otto Wene, Socrates Kypreos, etc. These models, e.g., MARKAL, preceding the mentioned TIAM, do explicitly take such energy system changes (and thus WTW effects) into account.

The question then is what is novel about this manuscript. The early studies did almost certainly not consider the CTG cycle, but the more recent ones might. See, e.g., GCAM at PNWL. This is something for the authors to find out. However, whether this potential novelty justifies publication in a Nature journal can be debated, particularly in light of the comparison in Fig.3.

The authors should also address why the comparison between BEV, HEV, PHEV, and FCEV matters. The second and third technology are merely transition technologies, whereas the last technology can almost certainly be ruled out as a realistic LDV option. The results would suggest that the energy transition should happen as soon as possible, given the comparatively low and declining CO₂ emissions from BEVs and the climate's dependence on cumulative CO₂ emissions. But doesn't this apply to all sectors and technologies?

In addition, the results heavily depend on the charging behaviour of consumers. The authors make some lip service on p.15, but should explore this shortcoming in more depth. When are most BEVs being charged? Looking at the literature might help. If it is during night at low wind conditions, fossil fuel-based electricity can quickly eliminate the identified benefits of BEVs.

Also, what is the specification of the four LDVs. What vehicle performance parameter has remained constant to ensure a like-for-like comparison?

Reviewer #3 (Remarks to the Author):

The core claim of the paper is that the paper aims to improve the accuracy of passenger vehicle carbon footprints under uncertain futures using pLCA method, Monte Carlo and global sensitivity analysis. The novelty lies in pLCA which will benefit to others in the community and wider field. The results are original and convincing somewhat (not strong though). The statistical analysis is robust. Other researchers should be able to reproduce the work.

There are some questions though:

1. The title is too general and not capturing or delivering accurately the content of the paper.
2. The structure of the paper is unusual, i.e. the results section is ahead of the methods section. This raised a lot of questions during reading the paper which are not necessary if it was structured in the conventional way. Authors should indicate this in the introduction section at least.
Also, there is no conclusion section.
3. Page 4, line 21, please define NDCs.
4. Page 5, figure 1, it would be helpful if the colours of the legends can have the same order as the graph (a).
5. Page 6, figure 2, the colours of future scenarios are similar with four vehicle types (at least BEV and very ambitious, both

green), causing confusions.

What is the purpose of four colours that represents vehicle types? Are they for triangle areas? Clarify.

6. Page 11, line 11, "Figure 3b shows that the reduction in HEV carbon footprint is relatively small.". Please articulate using numbers.

7. Page 12, line 7, "shows that vehicle size significantly impacts results across all vehicles". This is not the case for FECV.

8. Page 12, line 9, "heavier vehicles decrease their efficiency, consuming more electricity per kilometre". Inaccurate description, what efficiency?

9. Please spell out all the abbreviations at their first appearance, e.g. page 15, lines 23-24.

10. Page 17, second paragraph, Gasoline based hybrid vehicles are far outnumbered diesel based HEVs and PHEVs in the market. Can authors justify the decision please?

Communications Earth & Environment is committed to improving transparency in authorship. As part of our efforts in this direction, we are now requesting that all authors identified as 'corresponding author' create and link their Open Researcher and Contributor Identifier (ORCID) with their account on the Manuscript Tracking System prior to acceptance. ORCID helps the scientific community achieve unambiguous attribution of all scholarly contributions. You can create and link your ORCID from the home page of the Manuscript Tracking System by clicking on 'Modify my Springer Nature account' and following the instructions in the link below. Please also inform all co-authors that they can add their ORCIDs to their accounts and that they must do so prior to acceptance.

Version 1:

Decision Letter:

Dear Mr Simaitis,

Your manuscript titled "Battery electric vehicles consistently show the lowest carbon footprints among passenger cars across 1.5-3.0°C energy decarbonisation pathways" has now been seen by our reviewers, whose comments appear below. In light of their advice we are delighted to say that we are happy, in principle, to publish a suitably revised version in Communications Earth & Environment.

We therefore invite you to revise your paper one last time to address the remaining concerns of our reviewers. At the same time we ask that you edit your manuscript to comply with our format requirements and to maximise the accessibility and therefore the impact of your work.

EDITORIAL REQUESTS:

*****Please take care to match our formatting and policy requirements. We will check revised manuscript and return manuscripts that do not comply. Such requests will lead to delays. *****

SUBMISSION INFORMATION:

OPEN ACCESS:

Communications Earth & Environment is a fully open access journal. Articles are made freely accessible on publication. For further information about article processing charges, open access funding, and advice and support from Nature Research, please visit <https://www.nature.com/commsenv/open-access>

Link Redacted

Best regards,

Pallav Purohit, PhD
Editorial Board Member
Communications Earth & Environment

Martina Grecequet, PhD
Senior Editor,
Communications Earth & Environment
@CommsEarth

REVIEWERS' COMMENTS:

Reviewer #1 (Remarks to the Author):

I thank the authors for their thorough revision and for addressing all the reviewers' comments. They have significantly improved the explanation of the article's novelty in comparison to the existing literature. The inclusion of a clear justification for the use of TIAM-UCL scenarios is appreciated. The additional analysis in Section 2.4 provides valuable insights into regional variability. Overall, I am satisfied with their response.

Reviewer #2 (Remarks to the Author):

I don't mean to diminish the work by the authors, but I still have problems understanding the novelty of this submission. Based upon my understanding, the value added generated by this piece of work compared to the existing literature is incremental.

As for 2.4 in the reviewer response, the emissions mitigations that materialize there and then will obviously depend heavily on the modeller's input assumptions, whether they relate to technology/cost specifications, market penetration constraints, or discount rate. As such, the model does NOT remain "impartial".

With respect to 2.5, the authors have not responded to my concern regarding the temporal dimension of charging behaviour, e.g., day vs. night, and its impact on the results.

Reviewer #3 (Remarks to the Author):

The revised version well addressed the questions raised by the reviewers. The manuscript is ready to publish apart from a minor typo: line 29, page 4 in the revised version: Currently outperform other powertrains in most regions, where "in" is missing.

R	Comment	Author Response (direct references to manufacturing pages and line numbers correspond to the tracked changes version)
1.1	Comment 1: What is the actual novelty of the findings? From an application perspective, I'm unsure what truly distinguishes the findings of this paper from previously published research. While I acknowledge the methodological innovations presented in this work (mentioned above), the major conclusions appear to largely reiterate insights already established in prior studies. For example, the authors emphasize that "We show that incorporating future scenarios is crucial for determining comparative carbon footprints". While this point is undoubtedly important, it has already been addressed in earlier works, including the seminal study by Mendoza-Beltran et al. (DOI: 10.1111/jiec.12825), Cox et al. (https://doi.org/10.1016/j.apenergy.2020.115021), and Sacchi et al. (https://doi.org/10.1016/j.rser.2022.112475). For other findings, such as that "BEVs consistently show the lowest carbon footprints among passenger vehicles", the authors acknowledge that they align with previous studies. As a main difference with previous works, the authors highlight that "some studies have suggested that BEVs may have higher carbon footprints than combustion-dominant vehicles such as HEVs and ICEVs in regions with high-carbon electricity grids like China [6,9,22], India [6,8], and Eastern Europe [6,23]. In contrast, our study finds that current BEVs will generally retain the lowest carbon footprints even in these regions." This conclusion is noteworthy but not entirely convincing given the concerns raised in Comment #4 regarding the regional granularity of the analysis. Overall, I do not intend this comment to diminish the value of the authors' work, which I find undoubtedly commendable. However, the novelty of the findings remains unclear from the perspective of a reader familiar with the literature. What specific new insights does this study offer compared to the already existing body of literature? Moreover, how do the methodological advances introduced in the paper enabled these contributions? These are the two questions that came to mind after reading the paper.	[Major Revision] The study's novelty has been significantly highlighted with revisions throughout the Introduction (e.g. page 4 from line 18) and Discussion (page 17 from line 14) sections. Mendoza-Beltran et al. (2020) and Cox et al. (2020) only rely on two IMAGE scenarios and focus solely on electricity transformation while neglecting critical energy system shifts in fuels and hydrogen that is key for non-electric powertrains. They also do not adjust life cycle stages over time to account for variations in vehicle use impacts throughout its lifetime, reducing LCA accuracy. Sacchi et al. (2022) introduce time adjustments but still use only two REMIND scenarios and these are applied exclusively to the background system. Meanwhile, the foreground electricity mix - the primary driver of results - relies on a separate, single scenario per region, creating inconsistencies between background and foreground futures. Our study delivers the most methodologically consistent time-adjusted pLCA of passenger cars, integrating four distinct TIAM-UCL scenarios that fully capture electricity, fuel, and hydrogen transitions across all powertrains. We ensure a harmonised assessment by consistently applying IAM future scenarios to foreground and background systems. The broader scenario range enables a deeper exploration of future uncertainties, yielding more robust and reliable passenger car outcomes. Despite arriving at similar conclusions, our study significantly enhances the methodological rigor and reliability of outcomes for BEVs, which is still important in today's debates surrounding passenger cars which can still show hesitation for BEV adoption - e.g. EU's concession to still include potential for ICE to be sold on synthetic fuels [Regulation (EU) 2023/851], Toyota's strategy of hybrids and hydrogen, Porche's bet on ICE and synthetic fuels, or studies calling for more assessments of combustion-engine LCA results (Liu et al. 2024, DOI: 10.1016/j.jclepro.2024.141996).
1.2	Comment 2: On the energy scenarios from the TIAM-UCL model	[Minor Revision] The key advantage and justification for TIAM-UCL is added to the Introduction (page 5, from line 26)

	I highly appreciate the authors' effort in integrating energy scenarios from the TIAM-UCL model into prospective LCA and making them accessible to the community through premise. As the Results section starts with a description of these scenarios, it would be highly valuable to include a brief comparison highlighting the differences between the TIAM-UCL scenarios and those from REMIND and IMAGE, which are already available in premise. In short, what makes these scenarios more relevant, and what motivated the authors to undertake this effort instead of just using the existing REMIND/IMAGE scenarios? I believe providing this context would help readers better understand the important contribution of integrating these new scenarios.	A key advantage of TIAM-UCL is that it has a more detailed focus on energy systems, providing more technology detail and a more detailed representation of fossil-based systems, which is helpful in the electricity, fuel, and hydrogen scenarios. This is discussed in more detail in a preceding publication (Šimaitis et al. 2025, DOI: 10.1016/j.rser.2024.115298) that compares REMIND and IMAGE scenarios, while including this comparison in this work is out of scope.
1.3 Comment 3: Diesel and hydrogen cars without diesel and hydrogen? By examining the energy scenarios considered, it becomes evident that in certain cases the production of diesel and hydrogen is either extremely low or absent (e.g., hydrogen in the “No climate action” scenario before 2035 and diesel in the “Very Ambitious” scenario by 2050). In these situations, I was wondering whether it is realistic to include these vehicle type–scenario–year combinations in the pairwise comparison analysis in Section 2.2. I don't know to what extent these combinations influence the statistical analysis of the results presented in Figure 2, but their consideration may introduce highly implausible scenarios into the analysis. The authors could consider conducting a more detailed assessment to ensure only consistent and realistic combinations are included (similar to the approach discussed in Langkau et al.: https://doi.org/10.1007/s11367-023-02175-9).	[Rebuttal] We appreciate the reviewer's concern about depicting realistic future vehicle type–scenario–year combinations, mainly when diesel or hydrogen production is low. However, restricting our analysis to only certain combinations could unnecessarily limit the scope, exclude valuable edge cases, and introduce methodological inconsistencies, especially given that future scenarios are inherently explorative and uncertain, making it challenging to define what is truly realistic. Therefore, no changes are made. Our pLCA approach is inherently explorative. It employs Monte Carlo sampling to capture various scenario and powertrain uncertainties. Including a broad spectrum of combinations strengthens the robustness of our findings and ensures adaptability to future uncertainties.	
1.4 Comment 4: Potential limitations of the regional granularity I'd like to address one of the main conclusions of the article: “some studies have suggested that BEVs may have higher carbon footprints than combustion-dominant vehicles such as HEVs and ICEVs in regions with high-carbon electricity grids like China [6,9,22], India [6,8], and Eastern Europe [6,23]. In contrast, our study finds that current BEVs will generally retain the lowest carbon footprints even in these regions.” This statement seems to be an oversimplification. As the authors acknowledge in the discussion, “Another limitation is that TIAM-UCL aggregates regions into 16 major areas.” This level of aggregation may overlook the significant variability in electricity grids at the country or even subregional level. For instance, Sacchi et al. (2022) provide future carbon footprints at the country level, which is not captured in this study. This limitation raises questions	[Major Revision] The conclusions and limitations of regional granularity are further refined and expanded in the discussion section (page 19, line 9 onward). Additionally, to account for regional carbon intensity variations, we have added new trend analysis (Section 2.4) providing deeper insight into how regions not represented in TIAM-UCL, such as those with higher electricity grid intensities, may impact BEV-related conclusions using the breakeven mileage perspective and uncertainty. “some studies have suggested that BEVs may have higher carbon footprints than combustion-dominant vehicles such as HEVs and ICEVs in regions with high-carbon electricity grids like China [6,9,22] and India [6,8], and Eastern Europe [6,23]. In contrast, our study finds that current BEVs will generally retain the lowest carbon footprints even in these regions. The key distinction in our study is that by time-adjusting the pLCA with future decarbonisation trajectories, we project a much lower average carbon intensity for BEV electricity supply than typical LCA. This is especially true in high-	

	about the extent to which the regional granularity of this study (I acknowledge that this is a common issue when using IAMs scenarios in LCA) supports some of its broader conclusions. It is well understood that no electric vehicle will be charged using something such as a "Western Europe power grid." Heterogeneity exists within Western European countries or even regions within China or the US regarding their electricity mixes and potentially decarbonization objectives. Since regional variability is a core component of this work, I believe the authors should either refine the granularity of their analysis or adjust the discussion and ambition of their conclusions to better reflect this inherent limitation.	carbon regions like China, which can be expected to decarbonise over the next 10-20 years in all future trajectories. These conclusions focus on representing global and major regional future trends, but overgeneralisation to specific regions is cautioned. A key limitation of the consistent application of IAM scenarios to pLCA is the loss of regional resolution due to the aggregation of multiple regions, as seen in groupings such as Western Europe (WEU) or Eastern Europe (EEU). For instance, this aggregation may overlook edge cases in energy mix carbon intensities, such as the exceptionally high and low electricity mixes of Estonia and Norway, respectively [33]. Furthermore, while major regions such as the United States (US), China, and India do have their independent indicators in TIAM-UCL, significant provincial variations in energy mixes still introduces uncertainty [34]. The trend analysis on regional electricity carbon intensity illustrates how this factor impacts BEV breakeven mileage, accounting for potential variations that are not explicitly represented in specific TIAM-UCL regions. In regions with potentially higher carbon-intensive electricity, BEVs must be driven significantly more miles to compensate for their higher production emissions and achieve parity with HEVs. This adds uncertainty to their ability to achieve a meaningfully lower carbon footprint. However, even in the "no climate action" scenario, accounting for future decarbonisation could reduce this mileage, strengthening confidence in BEVs achieving meaningful carbon footprint reductions. On the other hand, TIAM-UCL is limited by its yearly temporal resolution, preventing it from capturing in-year variations in energy mixes, such as the hourly intermittency of renewables for charging³⁵. Therefore, applying these broader conclusions to specific contexts would ultimately need an enhanced geographical and temporal resolution of prospective energy mixes that cannot be captured by IAMs."
1.5	Comment 5: LCA inputs with greatest impacts In Section 2.4, the authors present a global sensitivity analysis showing the LCA inputs with the greatest impact on the carbon footprints. While the inputs considered for the GSA are those used in their modelling, I believe there is much more that could affect the carbon footprint and that at least deserve a qualitative assessment. Notably, the supply of the raw materials used in lithium-ion batteries, which have a large influence on the carbon footprint of the battery. For example, a recent paper by Peiseler et al. (https://doi.org/10.1038/s41467-024-54634-y) suggests that material sourcing is more relevant than production location. Overall, I believe the authors should discuss the relevance of the inputs in light of the fact that their analysis does not consider the influence of all possible inputs.	Minor Revision The potential influences of other inputs are added as a limitation (page 20, from line 24) "Although the study focused on advancing the prospective methodology and assessing the impact variation of prospective indicators, further work could explore the uncertainty in manufacturing impacts arising from variations in battery chemistry and raw material sourcing, such as lithium, nickel, and cobalt [39]"
1.6		Minor Revision The battery recycling credit depiction is clarified in Figure 3 caption.

	In Figure 3, the authors presents some avoided impacts attributed to recycling credits. The text suggests that these credits come from battery recycling. However, what about the recycling of the vehicle itself?	“Potential battery recycling credits outside the system boundary are shown for reference. However, these credits are not used to “reduce” the carbon footprint of vehicles such as the results in Figure 2.” The main LCA results are based on the cut-off method, meaning that the primary results in Figure 2 and the final scores do not include substitution or credits. Additionally, all vehicles are treated equally regarding non-battery components. Figure 3 specifically highlights potential recycling credits outside the system boundaries, as this is a common topic of discussion in the automotive industry to help representation. Further clarification on this has been provided, but this can also just be removed.
1.7	In the Discussion section, the authors mention the potential application of consequential LCA to account for system-wide changes resulting from a transition to alternative passenger car fleets. While the attributional vs. consequential LCA debate has long been a topic of discussion within the community, I wonder to what extent consequential LCA of entire sectors (and the concept of marginal suppliers) should be applied together with IAM scenarios. These scenarios already depict a specific situation for various sectors (often including the passenger transport sector) as a result of the simulation or optimization performed by the model. One may argue that only attributional LCA can be applied to assess the impacts at the fleet-level of the specific transportation sector as depicted by the IAM scenario. The point I'd like to make is that the authors raise an avenue for future research by suggesting application of consequential LCA alongside IAM scenarios, but they do not discuss the current state-of-the-art of this approach and the potential limitations.	[Minor Revision] The discussion between attributional and consequential approaches is removed, as IAM scenarios inherently account for system-wide changes. This topic has been identified as beyond scope of this work and would require a much more extensive level of discussion to appropriately reflect that cannot be covered in this work.
2.1	In making these claims, the authors don't seem to be aware of the vast body of energy modelling studies, starting after the first oil crisis at Oak Ridge National Lab, Pacific North-West National Lab (PNWL), IIASA, the IEA, the Paul Scherrer Institute, and other institutions. See publications by Jae Edmunds, Nebojsa Nakicenovic, Tom Kram, Clas-Otto Wene, Socrates Kypreos, etc. These models, e.g., MARKAL, preceding the mentioned TIAM, do explicitly take such energy system changes (and thus WTW effects) into account.	[Minor Revision] We appreciate the reviewer concern that a vast body of energy modelling studies are not extensively covered. However, the manuscript does not claim to be the first to model energy system changes explicitly and is not an energy system model (ESM) focussed study. The manuscript explicitly acknowledges that IAMs are widely recognized for assessing future energy transitions, with no claims to this be a novel approach. The study focuses on the use of IAMs, specifically TIAM-UCL, for prospective LCA within Premise - an emerging approach that enhances future LCA representation for technologies like passenger cars. TIAM-UCL is only the third IAM within Premise to enable this methodology. We have revised the following to be clearer about this distinction (page 4, line 4) “In recent years, prospective LCA (pLCA) using future scenarios from integrated assessment models (IAMs) has emerged that can help address these challenges [15,16]. IAMs are widely recognised

		climate mitigation tools for exploring future energy transitions through socioeconomic and cost-optimisation modelling [17], and are closely-related and complimentary to energy system models [18]. In particular, the introduction of "Premise"[19] has allowed IAM future scenarios to be integrated into ecoinvent [20] LCA database. Premise maps IAM variables to LCA activities and generates prospective versions of the LCA database by adjusting technologies' penetration share, efficiency and emission factors for a specific scenario and year, considering regionalised sectoral changes related to electricity, fuels, cement, steel, and more. This has enabled pLCA studies to explore the future environmental impacts of technologies such as batteries [21], hydrogen [22], and cement [23]. However, there have been few pLCA studies that utilise the IAM approach passenger vehicle comparisons [6,24,25]"
2.2	The question then is what is novel about this manuscript. The early studies did almost certainly not consider the CTG cycle, but the more recent ones might. See, e.g., GCAM at PNWL. This is something for the authors to find out. However, whether this potential novelty justifies publication in a Nature journal can be debated, particularly in light of the comparison in Fig.3.	[Rebuttal] Distinguishing the study's novelty has been substantially addressed with regards to comment 1.1 (see above). Moreover, comment 2.1. address that TIAM-UCL is only the third IAM to be linked with Premise and REMIND and IMAGE have already been discussed. Premise is currently the only harmonised methodology and framework for utilising IAM scenarios for LCA. Integrating LCA into IAMs or ESMs is not the scope of the paper. We are not able to address concerns with Figure 3 as these are not specified.
2.3	The authors should also address why the comparison between BEV, HEV, PHEV, and FCEV matters. The second and third technology are merely transition technologies, whereas the last technology can almost certainly ruled out as a realistic LDV option.	[Minor Revision] The inclusion of all powertrains is justified in the introduction (page 3, from line 11) due to ongoing debate over their carbon footprints and the continued investment by some manufacturers in alternatives, such as hybrid combustion technologies (see response to comment 1.1). While their role of hybrids as transitional technologies remains subjective, considering all powertrain types is essential, given future uncertainties in energy system transitions. Including all possible comparisons ensures the most comprehensive assessment accounting for all possibilities and future uncertainties (see response to comment 1.3). Although FCEVs are generally not expected to be a significant part of the technology mix (as noted on page 21, line 26), including them in the analysis remains valuable for considering future uncertainties and potential implementation in specific contexts. “Numerous life cycle assessment (LCA) studies indicate that BEVs generally have a lower life-cycle global warming potential (GWP, hereafter referred to as "carbon or carbon footprint") than ICEVs across most regions [3–5]. While BEVs have a higher embodied carbon due to energy-intensive battery production, this is outweighed by their efficient powertrain, which utilises electricity during operation that is less-carbon intensive compared to fossil-fuel supply and combustion in ICEVs [6,7]. However, in regions with high-carbon electricity grids, such as India or China, BEVs can have a higher carbon footprints than ICEVs [8,9]. Though, it is argued BEV use phase impacts will continue to decrease over time as electricity grids continue to decarbonise [10]. However, some arguments suggest that adopting low-carbon fuels could make combustion and hybrid powertrains competitive

		or even preferable [11–13]. In any case, the future energy mix, along with the magnitude and rate of decarbonisation, is uncertain but vital in determining passenger car carbon footprints.” “Although hydrogen fuel cell electric vehicles (FCEVs) have limited market shares due to early maturity and lacking hydrogen infrastructure, they are also included due to potential future interests [42]“
2.4	The results would suggest that the energy transition should happen as soon as possible, given the comparatively low and declining CO2 emissions from BEVs and the climate's dependence on cumulative CO2 emissions. But doesn't this apply to all sectors and technologies?	[Rebuttal] Pessimistic and optimistic decarbonization scenarios from TIAM-UCL are applied across multiple sectors (see Section 4.3), as for example, phasing out fossil fuels and scaling up solar and wind impact hydrogen production, liquid fuels, steel and more. Also, the significance of energy transition in achieving decarbonization varies across different sectors. For instance, in cement production, carbon capture and storage (CCS), combined with efficient heat recovery, remains effective even when the energy mix is not entirely green. Nonetheless, a high carbon price remains essential. The models remain impartial to vehicle selection, which is driven by cost, efficiency, and low-carbon fuel availability.
2.5	In addition, the results heavily depend on the charging behaviour of consumers. The authors make some lip service on p.15, but should explore this shortcoming in more depth. When are most BEVs being charged? Looking at the literature might help. If it is during night at low wind conditions, fossil fuel-based electricity can quickly eliminate the identified benefits of BEVs.	[Major Revision] The temporal limitations of this work is added to the discussion (page 19, line 9 onward), in addition a new analysis section supports part of this comment by investigating the effects of varying electricity carbon intensities (Section 2.4). Directly investigating charging behaviour and finer temporal granularity are beyond the study's scope, as we focus on global-level future scenarios using lifetime-averaged electricity mixes. This mid-to-long-term perspective contrasts with short-term consequential modelling, where the timing of charging would be more relevant. “These conclusions focus on representing global and major regional future trends, but overgeneralisation to specific regions is cautioned. A key limitation of the consistent application of IAM scenarios to pLCA is the loss of regional resolution due to the aggregation of multiple regions, as seen in groupings such as Western Europe (WEU) or Eastern Europe (EEU). For instance, this aggregation may overlook edge cases in energy mix carbon intensities, such as the exceptionally high and low electricity mixes of Estonia and Norway, respectively [33]. Furthermore, while major regions such as the United States (US), China, and India do have their independent indicators in TIAM-UCL, significant provincial variations in energy mixes still introduces uncertainty [34]. The trend analysis on regional electricity carbon intensity illustrates how this factor impacts BEV breakeven mileage, accounting for potential variations that are not explicitly represented in specific TIAM-UCL regions. In regions with potentially higher carbon-intensive electricity, BEVs must be driven significantly more miles to compensate for their higher production emissions and achieve parity with HEVs. This adds uncertainty to their ability to achieve a meaningfully lower carbon footprint. However, even in the “no climate action” scenario, accounting for future decarbonisation could reduce this mileage,

		strengthening confidence in BEVs achieving meaningful carbon footprint reductions. On the other hand, TIAM-UCL is limited by its yearly temporal resolution, preventing it from capturing in-year variations in energy mixes, such as the hourly intermittency of renewables for charging³⁵. Therefore, applying these broader conclusions to specific contexts would ultimately need an enhanced geographical and temporal resolution of prospective energy mixes that cannot be captured by IAMs.”
2.6	Also, what is the specification of the four LDVs. What vehicle performance parameter has remained constant to ensure a like-for-like comparison?	[Minor Revision] Section 4.2 (page 23, line 2) provides further clarification on key LDV specifications. The rest of the section highlights that the Calculator LCA model standardises key vehicle parameters (e.g., weight, energy consumption, efficiency) across powertrains. The Monte Carlo approach ensures consistent comparisons in mileage, region, size, and lifetime. “Vehicles of the same size class are designed to follow the WLTP driving cycle with the same speed profile, seating capacity, and roughly equivalent engine power (determined by the driving cycle) to ensure fair powertrain comparisons...”
3.1	The title is too general and not capturing or delivering accurately the content of the paper.	[Minor Revision] The title has been suggested to be changed to “Battery electric vehicles consistently show the lowest carbon footprints among passenger cars across 1.5-3.0°C energy decarbonisation pathways.” Other possible suggestions are given below. Future carbon footprints of passenger cars. Prospective life cycle assessment reveals that battery electric vehicles consistently show the lowest carbon footprints among passenger cars across 1.5-3.0°C decarbonisation pathways Prospective life cycle assessment explores 1.5-3.0°C energy decarbonisation scenarios to reveal that battery electric vehicles consistently show the lowest carbon footprints among passenger cars. Battery electric vehicles consistently show the lowest carbon footprints among passenger cars across 1.5-3.0°C energy decarbonisation pathways, now and in the future. Current and future battery electric vehicles consistently show the lowest carbon footprints among passenger cars across 1.5-3.0°C energy decarbonisation pathways.
3.2	The structure of the paper is unusual, i.e. the results section is ahead of the methods section. This raised a lot of questions during reading the paper which are not necessary if it was structured in the conventional way. Authors should	[Rebuttal] While we appreciate the reviewers concern surrounding the paper structure, this is also the journals required structure, so we are not able to change this.

	indicate this in the introduction section at least. Also, there is no conclusion section.	
3.3	Page 4, line 21, please define NDCs.	[Minor Revision] This has been corrected.
3.4	Page 5, figure 1, It would be helpful if the colours of the legends can have the same order as the graph (a).	[Rebuttal] The chart legends are ordered alphabetically and align with the data order within each chart. Fossil technologies are represented with darker colours, while renewable technologies use lighter shades for clear differentiation.
3.5	Page 6, figure 2, the colours of future scenarios are similar with four vehicle types (at least BEV and very ambitious, both green), causing confusions. What is the purpose of four colours that represents vehicle types? Are they for triangle areas? Clarify.	[Minor Revision] We acknowledge that some colours may overlap; however, variations in shading ensure that the points remain distinguishable. The reasoning for this, along with examples, is clarified further in the Figure 3 caption. “Figure 3: Comparative Monte Carlo results for the life-cycle carbon footprint of passenger vehicle powertrains, for four future scenarios. The production year is 2025 to show present-day vehicles that will be used and reach their end-of-life in 2035-2045. Results represent 5,000 unique configurations computed equivalently for BEV, PHEV, HEV, and FCEV. The mean points of each scenario are displayed in their respective colour and shape – defined in the legend as “Mean average for each future scenario”. For example, (a) represents comparable pairwise results for a BEV and HEV. Each point reflects a BEV vs. HEV result from the equivalent configuration of the same mileage, region, lifetime, future scenario, size etc. If a point is in the HEV area of the chart (shaded light yellow, represented in the “Greater impact area” legend), then the HEV had a higher carbon footprint than the BEV in that configuration. The opposite is true if a point is in the BEV area of the chart (shaded light green).”
3.6	Page 11, line 11, "Figure 3b shows that the reduction in HEV carbon footprint is relatively small.". Please articulate using numbers.	[Minor Revision] This articulation has been improved by adding the specific number percentages (page X, lines XX) “Similarly, Figure 4c illustrates that diesel does also decarbonise mainly via biodiesel uptake in the pLCA approach, as seen in the China (CHI) case by 41%. However, Figure 3a reveals that diesel supply represents a relatively minor contribution to the HEV carbon footprint since it is outweighed by the dominant exhaust emissions. Therefore, even when diesel decarbonisation is accounted for across 2025-2040, Figure 3b shows that HEV carbon footprint reduction is a relatively small 11%.”

3.7	Page 12, line 7, "shows that vehicle size significantly impacts results across all vehicles". This is not the case for FECV.	[Minor Revision] This has been adjusted on page 12 and lines 7. Figure 5a (left-hand column) shows that vehicle size significantly impacts results across all vehicles since it directly determines manufacturing and use-phase demands (except for FCEVs) For FCEVs, the future scenario is the most significant factor due to the "Very ambitious (1.5°C)" case."
3.8	Page 12, line 9, "heavier vehicles decrease their efficiency, consuming more electricity per kilometre". Inaccurate description: what efficiency?	[Minor Revision] This has been adjusted on page 12 and lines 7. "Also, heavier vehicles decrease their energy efficiency, consuming more electricity or fuel per kilometre."
3.9	Please spell out all the abbreviations at their first appearance, e.g. page 15, lines 23-24.	[Minor Revision] Checked and corrected throughout the manuscript to ensure abbreviations are spelled out once first appearing
3.10	Page 17, second paragraph, Gasoline based hybrid vehicles are far outnumbered diesel based HEVs and PHEVs in the market. Can authors justify the decision please?	[Minor Revision] The reason for this is has been justified in page 21 lines 15 "Diesel was chosen as the primary combustion fuel, rather than gasoline, due to its better technology representation in the TIAM-UCL future scenarios. For implications, diesel tends to have a lower carbon footprint compared to gasoline in the Calculator model, making it the more optimistic assumption for combustion powertrains than gasoline"

Responses to final revision

R	Comment	Author Response (direct references to manufacturing pages and line numbers correspond to the tracked changes version)
1	I thank the authors for their thorough revision and for addressing all the reviewers' comments. They have significantly improved the explanation of the article's novelty in comparison to the existing literature. The inclusion of a clear justification for the use of TIAM-UCL scenarios is appreciated. The additional analysis in Section 2.4 provides valuable insights into regional variability. Overall, I am satisfied with their response.	We are pleased that the revisions were satisfactory and thank the reviewer for their constructive feedback that has improved the manuscript.
2	I don't mean to diminish the work by the authors, but I still have problems understanding the novelty of this submission. Based upon my understanding, the value added generated by this piece of work compared to the existing literature is incremental. As for 2.4 in the reviewer response, the emissions mitigations that materialize there and then will obviously depend heavily on the modeller's input assumptions, whether they relate to technology/cost specifications, market penetration constraints, or discount rate. As such, the model does NOT remain "impartial". With respect to 2.5, the authors have not responded to my concern regarding the temporal dimension of charging behaviour, e.g., day vs. night, and its impact on the results.	These concerns were addressed in the previous response; therefore, no further changes are implemented. The study novelty indeed was previous a concern shared among all reviewers. Following the first revisions, this has been addressed to satisfaction of the other reviewers with major revisions to distinguish this study. While direct inputs from TIAM-UCL are beyond the scope of this study, the model operates impartially based on the assumptions provided by the modeller. Although the impartiality of these inputs can be debated - particularly in relation to the data and assumptions used – the essence of this issue is fundamentally acknowledged throughout the manuscript. The study emphasizes that the results are inherently sensitive to the choice of IAMs, and it highlights the need for further work to interpret outcomes by expanding the analysis to other IAMs. The authors directly responded to 2.5. To reiterate, the in-year temporal dimension on a daily or hourly basis is not possible to capture within the IAM in addition to not being in the paper scope. The study explicitly focuses on global, long-term future scenarios using lifetime-averaged electricity mixes. Nonetheless, we acknowledge the reviewers remark in the limitations section. Additionally, the temporal dimensions fundamentally refer to the variations of carbon-intensities, which is now captured in newly added analysis.
3	The revised version well addressed the questions raised by the reviewers. The manuscript is ready to publish apart from a minor typo: line 29, page 4 in the revised version: Currently outperform other powertrains in most regions, where "in" is missing.	We are pleased that the revisions were satisfactory and thank the reviewer for their constructive feedback that has improved the manuscript. We have fixed the stated typo.

Responses to first revision

R	Comment	Author Response (direct references to manufacturing pages and line numbers correspond to the tracked changes version)
1.1	Comment 1: What is the actual novelty of the findings? From an application perspective, I'm unsure what truly distinguishes the findings of this paper from previously published research. While I acknowledge the methodological innovations presented in this work (mentioned above), the major conclusions appear to largely reiterate insights already established in prior studies. For example, the authors emphasize that “We show that incorporating future scenarios is crucial for determining comparative carbon footprints”. While this point is undoubtedly important, it has already been addressed in earlier works, including the seminal study by Mendoza-Beltran et al. (DOI: 10.1111/jiec.12825), Cox et al. (https://doi.org/10.1016/j.apenergy.2020.115021), and Sacchi et al. (https://doi.org/10.1016/j.rser.2022.112475). For other findings, such as that “BEVs consistently show the lowest carbon footprints among passenger vehicles”, the authors acknowledge that they align with previous studies. As a main difference with previous works, the authors highlight that “some studies have suggested that BEVs may have higher carbon footprints than combustion-dominant vehicles such as HEVs and ICEVs in regions with high-carbon electricity grids like China [6,9,22], India [6,8], and Eastern Europe [6,23]. In contrast, our study finds that current BEVs will generally retain the lowest carbon footprints even in these regions.” This conclusion is noteworthy but not entirely convincing given the concerns raised in Comment #4 regarding the regional granularity of the analysis. Overall, I do not intend this comment to diminish the value of the authors' work, which I find undoubtedly commendable. However, the novelty of the findings remains unclear from the perspective of a reader familiar with the literature. What specific new insights does this study offer compared to the already existing body of literature? Moreover, how do the methodological advances introduced in the paper enabled these contributions? These are the two questions that came to mind after reading the paper.	[Major Revision] The study's novelty has been significantly highlighted with revisions throughout the Introduction (e.g. page 4 from line 18) and Discussion (page 17 from line 14) sections. Mendoza-Beltran et al. (2020) and Cox et al. (2020) only rely on two IMAGE scenarios and focus solely on electricity transformation while neglecting critical energy system shifts in fuels and hydrogen that is key for non-electric powertrains. They also do not adjust life cycle stages over time to account for variations in vehicle use impacts throughout its lifetime, reducing LCA accuracy. Sacchi et al. (2022) introduce time adjustments but still use only two REMIND scenarios and these are applied exclusively to the background system. Meanwhile, the foreground electricity mix - the primary driver of results - relies on a separate, single scenario per region, creating inconsistencies between background and foreground futures. Our study delivers the most methodologically consistent time-adjusted pLCA of passenger cars, integrating four distinct TIAM-UCL scenarios that fully capture electricity, fuel, and hydrogen transitions across all powertrains. We ensure a harmonised assessment by consistently applying IAM future scenarios to foreground and background systems. The broader scenario range enables a deeper exploration of future uncertainties, yielding more robust and reliable passenger car outcomes. Despite arriving at similar conclusions, our study significantly enhances the methodological rigor and reliability of outcomes for BEVs, which is still important in today's debates surrounding passenger cars which can still show hesitation for BEV adoption - e.g. EU's concession to still include potential for ICE to be sold on synthetic fuels [Regulation (EU) 2023/851], Toyota's strategy of hybrids and hydrogen, Porche's bet on ICE and synthetic fuels, or studies calling for more assessments of combustion-engine LCA results (Liu et al. 2024, DOI: 10.1016/j.jclepro.2024.141996).

1.2	Comment 2: On the energy scenarios from the TIAM-UCL model I highly appreciate the authors' effort in integrating energy scenarios from the TIAM-UCL model into prospective LCA and making them accessible to the community through premise. As the Results section starts with a description of these scenarios, it would be highly valuable to include a brief comparison highlighting the differences between the TIAM-UCL scenarios and those from REMIND and IMAGE, which are already available in premise. In short, what makes these scenarios more relevant, and what motivated the authors to undertake this effort instead of just using the existing REMIND/IMAGE scenarios? I believe providing this context would help readers better understand the important contribution of integrating these new scenarios.	[Minor Revision] The key advantage and justification for TIAM-UCL is added to the Introduction (page 5, from line 26) A key advantage of TIAM-UCL is that it has a more detailed focus on energy systems, providing more technology detail and a more detailed representation of fossil-based systems, which is helpful in the electricity, fuel, and hydrogen scenarios. This is discussed in more detail in a preceding publication (Šimaitis et al. 2025, DOI: 10.1016/j.rser.2024.115298) that compares REMIND and IMAGE scenarios, while including this comparison in this work is out of scope.
1.3	Comment 3: Diesel and hydrogen cars without diesel and hydrogen? By examining the energy scenarios considered, it becomes evident that in certain cases the production of diesel and hydrogen is either extremely low or absent (e.g., hydrogen in the "No climate action" scenario before 2035 and diesel in the "Very Ambitious" scenario by 2050). In these situations, I was wondering whether it is realistic to include these vehicle type–scenario–year combinations in the pairwise comparison analysis in Section 2.2. I don't know to what extent these combinations influence the statistical analysis of the results presented in Figure 2, but their consideration may introduce highly implausible scenarios into the analysis. The authors could consider conducting a more detailed assessment to ensure only consistent and realistic combinations are included (similar to the approach discussed in Langkau et al.: https://doi.org/10.1007/s11367-023-02175-9).	[Rebuttal] We appreciate the reviewer's concern about depicting realistic future vehicle type–scenario–year combinations, mainly when diesel or hydrogen production is low. However, restricting our analysis to only certain combinations could unnecessarily limit the scope, exclude valuable edge cases, and introduce methodological inconsistencies, especially given that future scenarios are inherently explorative and uncertain, making it challenging to define what is truly realistic. Therefore, no changes are made. Our pLCA approach is inherently explorative. It employs Monte Carlo sampling to capture various scenario and powertrain uncertainties. Including a broad spectrum of combinations strengthens the robustness of our findings and ensures adaptability to future uncertainties.
1.4	Comment 4: Potential limitations of the regional granularity I'd like to address one of the main conclusions of the article: "some studies have suggested that BEVs may have higher carbon footprints than combustion-dominant vehicles such as HEVs and ICEVs in regions with high-carbon electricity grids like China [6,9,22], India [6,8], and Eastern Europe [6,23]. In contrast, our study finds that current BEVs will generally retain the lowest carbon footprints even in these regions." This statement seems to be an oversimplification. As the authors acknowledge in the discussion, "Another limitation is that TIAM-UCL aggregates regions into 16 major areas." This level of aggregation may overlook the significant variability in electricity grids at the country or even subregional	[Major Revision] The conclusions and limitations of regional granularity are further refined and expanded in the discussion section (page 19, line 9 onward). Additionally, to account for regional carbon intensity variations, we have added new trend analysis (Section 2.4) providing deeper insight into how regions not represented in TIAM-UCL, such as those with higher electricity grid intensities, may impact BEV-related conclusions using the breakeven mileage perspective and uncertainty. "some studies have suggested that BEVs may have higher carbon footprints than combustion-dominant vehicles such as HEVs and ICEVs in regions with high-carbon electricity grids like China [6,9,22] and India [6,8], and Eastern Europe [6,23]. In contrast, our study finds that current BEVs will generally retain the lowest carbon footprints even in these regions. The key distinction in our study is

	level. For instance, Sacchi et al. (2022) provide future carbon footprints at the country level, which is not captured in this study. This limitation raises questions about the extent to which the regional granularity of this study (I acknowledge that this is a common issue when using IAMs scenarios in LCA) supports some of its broader conclusions. It is well understood that no electric vehicle will be charged using something such as a "Western Europe power grid." Heterogeneity exists within Western European countries or even regions within China or the US regarding their electricity mixes and potentially decarbonization objectives. Since regional variability is a core component of this work, I believe the authors should either refine the granularity of their analysis or adjust the discussion and ambition of their conclusions to better reflect this inherent limitation.	that by time-adjusting the pLCA with future decarbonisation trajectories, we project a much lower average carbon intensity for BEV electricity supply than typical LCA. This is especially true in high-carbon regions like China, which can be expected to decarbonise over the next 10-20 years in all future trajectories. These conclusions focus on representing global and major regional future trends, but overgeneralisation to specific regions is cautioned. A key limitation of the consistent application of IAM scenarios to pLCA is the loss of regional resolution due to the aggregation of multiple regions, as seen in groupings such as Western Europe (WEU) or Eastern Europe (EEU). For instance, this aggregation may overlook edge cases in energy mix carbon intensities, such as the exceptionally high and low electricity mixes of Estonia and Norway, respectively [33]. Furthermore, while major regions such as the United States (US), China, and India do have their independent indicators in TIAM-UCL, significant provincial variations in energy mixes still introduces uncertainty [34]. The trend analysis on regional electricity carbon intensity illustrates how this factor impacts BEV breakeven mileage, accounting for potential variations that are not explicitly represented in specific TIAM-UCL regions. In regions with potentially higher carbon-intensive electricity, BEVs must be driven significantly more miles to compensate for their higher production emissions and achieve parity with HEVs. This adds uncertainty to their ability to achieve a meaningfully lower carbon footprint. However, even in the "no climate action" scenario, accounting for future decarbonisation could reduce this mileage, strengthening confidence in BEVs achieving meaningful carbon footprint reductions. On the other hand, TIAM-UCL is limited by its yearly temporal resolution, preventing it from capturing in-year variations in energy mixes, such as the hourly intermittency of renewables for charging³⁵. Therefore, applying these broader conclusions to specific contexts would ultimately need an enhanced geographical and temporal resolution of prospective energy mixes that cannot be captured by IAMs."
1.5	Comment 5: LCA inputs with greatest impacts In Section 2.4, the authors present a global sensitivity analysis showing the LCA inputs with the greatest impact on the carbon footprints. While the inputs considered for the GSA are those used in their modelling, I believe there is much more that could affect the carbon footprint and that at least deserve a qualitative assessment. Notably, the supply of the raw materials used in lithium-ion batteries, which have a large influence on the carbon footprint of the battery. For example, a recent paper by Peiseler et al. (https://doi.org/10.1038/s41467-024-54634-y) suggests that material sourcing is more relevant than production location. Overall, I believe the authors should discuss the relevant of the inputs in light of the fact that their analysis do not consider the influence of all possible inputs.	[Minor Revision] The potential influences of other inputs are added as a limitation (page 20, from line 24) "Although the study focused on advancing the prospective methodology and assessing the impact variation of prospective indicators, further work could explore the uncertainty in manufacturing impacts arising from variations in battery chemistry and raw material sourcing, such as lithium, nickel, and cobalt [39]"

1.6	In Figure 3, the authors presents some avoided impacts attributed to recycling credits. The text suggests that these credits come from battery recycling. However, what about the recycling of the vehicle itself?	[Minor Revision] The battery recycling credit depiction is clarified in Figure 3 caption. “Potential battery recycling credits outside the system boundary are shown for reference. However, these credits are not used to “reduce” the carbon footprint of vehicles such as the results in Figure 2.” The main LCA results are based on the cut-off method, meaning that the primary results in Figure 2 and the final scores do not include substitution or credits. Additionally, all vehicles are treated equally regarding non-battery components. Figure 3 specifically highlights potential recycling credits outside the system boundaries, as this is a common topic of discussion in the automotive industry to help representation. Further clarification on this has been provided, but this can also just be removed.
1.7	In the Discussion section, the authors mention the potential application of consequential LCA to account for system-wide changes resulting from a transition to alternative passenger car fleets. While the attributional vs. consequential LCA debate has long been a topic of discussion within the community, I wonder to what extent consequential LCA of entire sectors (and the concept of marginal suppliers) should be applied together with IAM scenarios. These scenarios already depict a specific situation for various sectors (often including the passenger transport sector) as a result of the simulation or optimization performed by the model. One may argue that only attributional LCA can be applied to assess the impacts at the fleet-level of the specific transportation sector as depicted by the IAM scenario. The point I’d like to make is that the authors raise an avenue for future research by suggesting application of consequential LCA alongside IAM scenarios, but they do not discuss the current state-of-the-art of this approach and the potential limitations.	[Minor Revision] The discussion between attributional and consequential approaches is removed, as IAM scenarios inherently account for system-wide changes. This topic has been identified as beyond scope of this work and would require a much more extensive level of discussion to appropriately reflect that cannot be covered in this work.
2.1	In making these claims, the authors don't seem to be aware of the vast body of energy modelling studies, starting after the first oil crisis at Oak Ridge National Lab, Pacific North-West National Lab (PNWL), IIASA, the IEA, the Paul Scherrer Institute, and other institutions. See publications by Jae Edmunds, Nebojsa Nakicenovic, Tom Kram, Clas-Otto Wene, Socrates Kypreos, etc. These models, e.g., MARKAL, preceding the mentioned TIAM, do explicitly take such energy system changes (and thus WTW effects) into account.	[Minor Revision] We appreciate the reviewer concern that a vast body of energy modelling studies are not extensively covered. However, the manuscript does not claim to be the first to model energy system changes explicitly and is not an energy system model (ESM) focussed study. The manuscript explicitly acknowledges that IAMs are widely recognized for assessing future energy transitions, with no claims to this be a novel approach. The study focuses on the use of IAMs, specifically TIAM-UCL, for prospective LCA within Premise - an emerging approach that enhances future LCA representation for technologies like passenger cars. TIAM-UCL is only the third IAM within Premise to enable this methodology. We have revised the following to be clearer about this distinction (page 4, line 4)

		“In recent years, prospective LCA (pLCA) using future scenarios from integrated assessment models (IAMs) has emerged that can help address these challenges [15,16]. IAMs are widely recognised climate mitigation tools for exploring future energy transitions through socioeconomic and cost-optimisation modelling [17], and are closely-related and complimentary to energy system models [18]. In particular, the introduction of "Premise"[19] has allowed IAM future scenarios to be integrated into ecoinvent [20] LCA database. Premise maps IAM variables to LCA activities and generates prospective versions of the LCA database by adjusting technologies’ penetration share, efficiency and emission factors for a specific scenario and year, considering regionalised sectoral changes related to electricity, fuels, cement, steel, and more. This has enabled pLCA studies to explore the future environmental impacts of technologies such as batteries [21], hydrogen [22], and cement [23]. However, there have been few pLCA studies that utilise the IAM approach passenger vehicle comparisons [6,24,25]”
2.2	The question then is what is novel about this manuscript. The early studies did almost certainly not consider the CTG cycle, but the more recent ones might. See, e.g., GCAM at PNWL. This is something for the authors to find out. However, whether this potential novelty justifies publication in a Nature journal can be debated, particularly in light of the comparison in Fig.3.	[Rebuttal] Distinguishing the study’s novelty has been substantially addressed with regards to comment 1.1 (see above). Moreover, comment 2.1. address that TIAM-UCL is only the third IAM to be linked with Premise and REMIND and IMAGE have already been discussed. Premise is currently the only harmonised methodology and framework for utilising IAM scenarios for LCA. Integrating LCA into IAMs or ESMs is not the scope of the paper. We are not able to address concerns with Figure 3 as these are not specified.
2.3	The authors should also address why the comparison between BEV, HEV, PHEV, and FCEV matters. The second and third technology are merely transition technologies, whereas the last technology can almost certainly ruled out as a realistic LDV option.	[Minor Revision] The inclusion of all powertrains is justified in the introduction (page 3, from line 11) due to ongoing debate over their carbon footprints and the continued investment by some manufacturers in alternatives, such as hybrid combustion technologies (see response to comment 1.1). While their role of hybrids as transitional technologies remains subjective, considering all powertrain types is essential, given future uncertainties in energy system transitions. Including all possible comparisons ensures the most comprehensive assessment accounting for all possibilities and future uncertainties (see response to comment 1.3). Although FCEVs are generally not expected to be a significant part of the technology mix (as noted on page 21, line 26), including them in the analysis remains valuable for considering future uncertainties and potential implementation in specific contexts. “Numerous life cycle assessment (LCA) studies indicate that BEVs generally have a lower life-cycle global warming potential (GWP, hereafter referred to as "carbon or carbon footprint") than ICEVs across most regions [3–5]. While BEVs have a higher embodied carbon due to energy-intensive battery production, this is outweighed by their efficient powertrain, which utilises electricity during operation that is less-carbon intensive compared to fossil-fuel supply and combustion in ICEVs [6,7]. However, in regions with high-carbon electricity grids, such as India or China, BEVs can have a higher carbon footprints than ICEVs [8,9]. Though, it is argued BEV use phase impacts will continue to

		decrease over time as electricity grids continue to decarbonise [10]. However, some arguments suggest that adopting low-carbon fuels could make combustion and hybrid powertrains competitive or even preferable [11–13]. In any case, the future energy mix, along with the magnitude and rate of decarbonisation, is uncertain but vital in determining passenger car carbon footprints.” “Although hydrogen fuel cell electric vehicles (FCEVs) have limited market shares due to early maturity and lacking hydrogen infrastructure, they are also included due to potential future interests [42]“
2.4	The results would suggest that the energy transition should happen as soon as possible, given the comparatively low and declining CO2 emissions from BEVs and the climate's dependence on cumulative CO2 emissions. But doesn't this apply to all sectors and technologies?	[Rebuttal] Pessimistic and optimistic decarbonization scenarios from TIAM-UCL are applied across multiple sectors (see Section 4.3), as for example, phasing out fossil fuels and scaling up solar and wind impact hydrogen production, liquid fuels, steel and more. Also, the significance of energy transition in achieving decarbonization varies across different sectors. For instance, in cement production, carbon capture and storage (CCS), combined with efficient heat recovery, remains effective even when the energy mix is not entirely green. Nonetheless, a high carbon price remains essential. The models remain impartial to vehicle selection, which is driven by cost, efficiency, and low-carbon fuel availability.
2.5	In addition, the results heavily depend on the charging behaviour of consumers. The authors make some lip service on p.15, but should explore this shortcoming in more depth. When are most BEVs being charged? Looking at the literature might help. If it is during night at low wind conditions, fossil fuel-based electricity can quickly eliminate the identified benefits of BEVs.	[Major Revision] The temporal limitations of this work is added to the discussion (page 19, line 9 onward), in addition a new analysis section supports part of this comment by investigating the effects of varying electricity carbon intensities (Section 2.4). Directly investigating charging behaviour and finer temporal granularity are beyond the study's scope, as we focus on global-level future scenarios using lifetime-averaged electricity mixes. This mid-to-long-term perspective contrasts with short-term consequential modelling, where the timing of charging would be more relevant. “These conclusions focus on representing global and major regional future trends, but overgeneralisation to specific regions is cautioned. A key limitation of the consistent application of IAM scenarios to pLCA is the loss of regional resolution due to the aggregation of multiple regions, as seen in groupings such as Western Europe (WEU) or Eastern Europe (EEU). For instance, this aggregation may overlook edge cases in energy mix carbon intensities, such as the exceptionally high and low electricity mixes of Estonia and Norway, respectively [33]. Furthermore, while major regions such as the United States (US), China, and India do have their independent indicators in TIAM-UCL, significant provincial variations in energy mixes still introduces uncertainty [34]. The trend analysis on regional electricity carbon intensity illustrates how this factor impacts BEV breakeven mileage, accounting for potential variations that are not explicitly represented in specific TIAM-UCL regions. In regions with potentially higher carbon-intensive electricity, BEVs must be driven significantly more miles to compensate for their higher production emissions and achieve parity with HEVs. This adds

		uncertainty to their ability to achieve a meaningfully lower carbon footprint. However, even in the “no climate action” scenario, accounting for future decarbonisation could reduce this mileage, strengthening confidence in BEVs achieving meaningful carbon footprint reductions. On the other hand, TIAM-UCL is limited by its yearly temporal resolution, preventing it from capturing in-year variations in energy mixes, such as the hourly intermittency of renewables for charging³⁵. Therefore, applying these broader conclusions to specific contexts would ultimately need an enhanced geographical and temporal resolution of prospective energy mixes that cannot be captured by IAMs.”
2.6	Also, what is the specification of the four LDVs. What vehicle performance parameter has remained constant to ensure a like-for-like comparison?	[Minor Revision] Section 4.2 (page 23, line 2) provides further clarification on key LDV specifications. The rest of the section highlights that the Carculator LCA model standardises key vehicle parameters (e.g., weight, energy consumption, efficiency) across powertrains. The Monte Carlo approach ensures consistent comparisons in mileage, region, size, and lifetime. “Vehicles of the same size class are designed to follow the WLTP driving cycle with the same speed profile, seating capacity, and roughly equivalent engine power (determined by the driving cycle) to ensure fair powertrain comparisons...”
3.1	The title is too general and not capturing or delivering accurately the content of the paper.	[Minor Revision] The title has been suggested to be changed to “Battery electric vehicles consistently show the lowest carbon footprints among passenger cars across 1.5-3.0°C energy decarbonisation pathways.” Other possible suggestions are given below. Future carbon footprints of passenger cars. Prospective life cycle assessment reveals that battery electric vehicles consistently show the lowest carbon footprints among passenger cars across 1.5-3.0°C decarbonisation pathways Prospective life cycle assessment explores 1.5-3.0°C energy decarbonisation scenarios to reveal that battery electric vehicles consistently show the lowest carbon footprints among passenger cars. Battery electric vehicles consistently show the lowest carbon footprints among passenger cars across 1.5-3.0°C energy decarbonisation pathways, now and in the future. Current and future battery electric vehicles consistently show the lowest carbon footprints among passenger cars across 1.5-3.0°C energy decarbonisation pathways.
3.2	The structure of the paper is unusual, i.e. the results section is ahead of the methods section. This raised a lot of questions during reading the paper which	[Rebuttal] While we appreciate the reviewers concern surrounding the paper structure, this is also the journals required structure, so we are not able to change this.

	are not necessary if it was structured in the conventional way. Authors should indicate this in the introduction section at least. Also, there is no conclusion section.	
3.3	Page 4, line 21, please define NDCs.	[Minor Revision] This has been corrected.
3.4	Page 5, figure 1, It would be helpful if the colours of the legends can have the same order as the graph (a).	[Rebuttal] The chart legends are ordered alphabetically and align with the data order within each chart. Fossil technologies are represented with darker colours, while renewable technologies use lighter shades for clear differentiation.
3.5	Page 6, figure 2, the colours of future scenarios are similar with four vehicle types (at least BEV and very ambitious, both green), causing confusions. What is the purpose of four colours that represents vehicle types? Are they for triangle areas? Clarify.	[Minor Revision] We acknowledge that some colours may overlap; however, variations in shading ensure that the points remain distinguishable. The reasoning for this, along with examples, is clarified further in the Figure 3 caption. “Figure 3: Comparative Monte Carlo results for the life-cycle carbon footprint of passenger vehicle powertrains, for four future scenarios. The production year is 2025 to show present-day vehicles that will be used and reach their end-of-life in 2035-2045. Results represent 5,000 unique configurations computed equivalently for BEV, PHEV, HEV, and FCEV. The mean points of each scenario are displayed in their respective colour and shape – defined in the legend as “Mean average for each future scenario”. For example, (a) represents comparable pairwise results for a BEV and HEV. Each point reflects a BEV vs. HEV result from the equivalent configuration of the same mileage, region, lifetime, future scenario, size etc. If a point is in the HEV area of the chart (shaded light yellow, represented in the “Greater impact area” legend), then the HEV had a higher carbon footprint than the BEV in that configuration. The opposite is true if a point is in the BEV area of the chart (shaded light green).”
3.6	Page 11, line 11, "Figure 3b shows that the reduction in HEV carbon footprint is relatively small.". Please articulate using numbers.	[Minor Revision] This articulation has been improved by adding the specific number percentages (page X, lines XX) “Similarly, Figure 4c illustrates that diesel does also decarbonise mainly via biodiesel uptake in the pLCA approach, as seen in the China (CHI) case by 41%. However, Figure 3a reveals that diesel supply represents a relatively minor contribution to the HEV carbon footprint since it is outweighed by the dominant exhaust emissions. Therefore, even when diesel decarbonisation is accounted for across 2025-2040, Figure 3b shows that HEV carbon footprint reduction is a relatively small 11%.”

3.7	Page 12, line 7, "shows that vehicle size significantly impacts results across all vehicles". This is not the case for FECV.	[Minor Revision] This has been adjusted on page 12 and lines 7. Figure 5a (left-hand column) shows that vehicle size significantly impacts results across all vehicles since it directly determines manufacturing and use-phase demands (except for FCEVs) For FCEVs, the future scenario is the most significant factor due to the "Very ambitious (1.5°C)" case."
3.8	Page 12, line 9, "heavier vehicles decrease their efficiency, consuming more electricity per kilometre". Inaccurate description: what efficiency?	[Minor Revision] This has been adjusted on page 12 and lines 7. "Also, heavier vehicles decrease their energy efficiency, consuming more electricity or fuel per kilometre. "
3.9	Please spell out all the abbreviations at their first appearance, e.g. page 15, lines 23-24.	[Minor Revision] Checked and corrected throughout the manuscript to ensure abbreviations are spelled out once first appearing
3.10	Page 17, second paragraph, Gasoline based hybrid vehicles are far outnumbered diesel based HEVs and PHEVs in the market. Can authors justify the decision please?	[Minor Revision] The reason for this is has been justified in page 21 lines 15 "Diesel was chosen as the primary combustion fuel, rather than gasoline, due to its better technology representation in the TIAM-UCL future scenarios. For implications, diesel tends to have a lower carbon footprint compared to gasoline in the Calculator model, making it the more optimistic assumption for combustion powertrains than gasoline"